JCB | Journal of Cell Biology

# Autoinhibition of Cnn binding to γ-TuRCs prevents ectopic microtubule nucleation and cell division defects

Corinne A. Tovey[1,2]⬤, Chisato Tsuji[1]⬤, Alice Egerton[1]⬤, Fred Bernard[2], Antoine Guichet[2]⬤, Marc de la Roche[3]⬤, and Paul T. Conduit[1,2]⬤

γ-Tubulin ring complexes (γ-TuRCs) nucleate microtubules. They are recruited to centrosomes in dividing cells via binding to N-terminal CM1 domains within γ-TuRC–tethering proteins, including *Drosophila* Centrosomin (Cnn). Binding promotes microtubule nucleation and is restricted to centrosomes in dividing cells, but the mechanism regulating binding remains unknown. Here, we identify an extreme N-terminal CM1 autoinhibition (CAI) domain found specifically within the centrosomal isoform of Cnn (Cnn-C) that inhibits γ-TuRC binding. Robust binding occurs after removal of the CAI domain or with the addition of phosphomimetic mutations, suggesting that phosphorylation helps relieve inhibition. We show that regulation of Cnn binding to γ-TuRCs is isoform specific and that misregulation of binding can result in ectopic cytosolic microtubules and major defects during cell division. We also find that human CDK5RAP2 is autoinhibited from binding γ-TuRCs, suggesting conservation across species. Overall, our results shed light on how and why CM1 domain binding to γ-TuRCs is regulated.

## Introduction

Microtubules are organized into specialized arrays crucial for cell function, such as the mitotic spindle. Correct array assembly relies in part on the spatiotemporal regulation of microtubule formation, and this is achieved by restricting microtubule formation and organization to microtubule organizing centers (MTOCs), such as the centrosome during mitosis (Tillery et al., 2018; Sanchez and Feldman, 2017; Petry and Vale, 2015).

The common link between most MTOCs is the presence of multiprotein γ-tubulin ring complexes (γ-TuRCs), which template and catalyze the kinetically unfavorable process of microtubule nucleation (Kollman et al., 2011; Teixidó-Travesa et al., 2012; Lin et al., 2015; Tovey and Conduit, 2018; Farache et al., 2018). γ-TuRCs are recruited to MTOCs by γ-TuRC–tethering proteins that directly link γ-TuRCs to the MTOC. γ-TuRCs contain 14 γ-tubulin molecules held in a single-turn helical conformation by laterally associating γ-tubulin complex proteins (GCPs). Each γ-tubulin molecule binds directly to an α-/β-tubulin dimer to promote new microtubule assembly. γ-TuRCs have a low activity within the cytosol but are thought to be activated after recruitment to MTOCs. In this model, the controlled recruitment and activation of γ-TuRCs enables the spatiotemporal control of microtubule nucleation and array formation. Recent structural studies have shown that γ-TuRCs purified from the cytosol of HeLa cells and *Xenopus* eggs are in a

semi-open conformation in which γ-tubulin molecules do not perfectly match the geometry of a 13-protofilament microtubule (Consolati et al., 2020; Liu et al., 2020; Wieczorek et al., 2020). This is also observed in recombinantly generated human γ-TuRCs (Zimmermann et al., 2020; Wieczorek et al., 2021; Würtz et al., 2021). A conformational change into a fully closed ring that matches microtubule geometry is expected to increase the nucleation capacity of the γ-TuRC. This is in agreement with studies in budding yeast showing conformational differences between γ-TuRC–like structures formed in vitro and γ-TuRCs bound to microtubules in vivo, and where artificial closure of γ-TuRCs increases microtubule nucleation capacity (Kollman et al., 2015).

How activation via an open-to-closed conformation change occurs is currently unclear, but various factors have been reported to increase nucleation capacity. γ-TuRCs purified from *Xenopus* egg extract nucleate much more efficiently after the addition of the tumor overexpressed gene (TOG) domain protein XMAP215 (Thawani et al., 2020). TOG domain family members mediate α-/β-tubulin addition via their TOG domains (Nithianantham et al., 2018), bind directly to γ-tubulin, and function in microtubule nucleation in vitro and in vivo (Wieczorek et al., 2015; Roostalu et al., 2015; Thawani et al., 2018; Flor-Parra et al., 2018; Gunzelmann et al., 2018). Single-molecule

................................................................................................................................................................................................................................
[1]Department of Zoology, University of Cambridge, Cambridge, UK;   [2]Université de Paris, Centre National de la Recherche Scientifique, Institut Jacques Monod, Paris, France;   [3]Department of Biochemistry, University of Cambridge, Cambridge, UK.

Correspondence to Paul T. Conduit: paul.conduit@ijm.fr.

experiments combined with modeling suggest that XMAP215 indirectly promotes the open-to-closed conformation change of purified γ-TuRCs by increasing the chance of protofilament formation, with lateral contacts between protofilaments promoting γ-TuRC closure (Thawani et al., 2020). While this is an attractive model, evidence suggests that activation can occur in different ways and may be context specific. Phosphorylation of γ-TuRCs by Aurora A around mitotic chromatin increases γ-TuRC activity (Pinyol et al., 2013; Scrofani et al., 2015), as does the addition of Nucleoside Diphosphate Kinase 7 (NME7) kinase in vitro (Liu et al., 2014). γ-TuRC activity is also increased after binding of the Augmin complex (Tariq et al., 2020), which tethers γ-TuRCs to other microtubules.

Another well-documented potential γ-TuRC activator is the Centrosomin motif 1 (CM1) domain, which is conserved in γ-TuRC–tethering proteins across eukaryotes (Sawin et al., 2004; Zhang and Megraw, 2007; Lin et al., 2014). Addition of protein fragments containing the CM1 domain increases the nucleation capacity of γ-TuRCs purified from human cells (Choi et al., 2010; Muroyama et al., 2016), although the degree of this activity change is much lower or absent when using γ-TuRCs purified from *Xenopus* eggs (Liu et al., 2020; Thawani et al., 2020). Expression of CM1 domain fragments within human cells leads to ectopic cytosolic microtubule nucleation, and this is dependent on CM1 binding to γ-TuRCs (Choi et al., 2010; Hanafusa et al., 2015; Cota et al., 2017). In fission yeast, expression of CM1 domain fragments also results in cytosolic microtubule nucleation (Lynch et al., 2014), and in *Xenopus* addition of CM1 domain fragments increases microtubule aster formation within egg extracts supplemented with activated Ran (Liu et al., 2020). In budding yeast, CM1 domain binding appears to move γ-tubulin molecules into a better position for nucleation (Brilot et al., 2021). While large global structural changes were not observed in mammalian γ-TuRCs bound by the CM1 domain (Liu et al., 2020; Wieczorek et al., 2020), local structural changes can be observed, suggesting that more global changes could, in theory, occur with a higher stoichiometry of binding (Brilot et al., 2021).

Given that CM1 domain binding leads to microtubule nucleation, binding is likely spatiotemporally controlled, particularly during cell division. This idea is consistent with results from numerous mass spectrometry experiments showing that γ-TuRCs do not readily associate with CM1 domain proteins within the cytosol (Oegema et al., 1999; Choi et al., 2010; Hutchins et al., 2010; Teixidó-Travesa et al., 2012; Thawani et al., 2018; Liu et al., 2020; Wieczorek et al., 2020; Consolati et al., 2020). Binding of the human and *Caenorhabditis elegans* CM1 domain proteins, CDK5RAP2 and SPD-5, to γ-TuRCs involves phosphorylation (Hanafusa et al., 2015; Ohta et al., 2021), which could be a means to spatiotemporally control binding. Nevertheless, whether phosphorylation directly promotes binding to γ-TuRCs or regulates binding in a different way remains unclear.

*Drosophila* Centrosomin (Cnn) is the only reported CM1 domain protein in *Drosophila* but is a multi-isoform gene with all isoforms containing the CM1 domain (Eisman et al., 2009). The centrosomal isoform (Cnn-C) has a dual role in both recruiting

γ-TuRCs to centrosomes (Zhang and Megraw, 2007; Conduit et al., 2014b) and forming a centrosome scaffold that supports mitotic pericentriolar material assembly (Conduit et al., 2014a; Feng et al., 2017). Phosphorylation of a central phospho-regulated multimerization (PReM) domain specifically at centrosomes promotes interactions between the PReM and C-terminal CM2 domains and drives the oligomerization of Cnn-C molecules into a scaffold-like structure that helps recruit other centrosomal proteins (Conduit et al., 2014a; Feng et al., 2017). Testes-specific Cnn-T isoforms have mitochondrial localization domains instead of the PReM and CM2 domains and recruit γ-TuRCs to mitochondria in sperm cells (Chen et al., 2017). Cnn-C and Cnn-T also vary in their extreme N-terminal regions, upstream of the CM1 domain, with Cnn-C containing a longer sequence.

Here, we show that the longer, extreme N-terminal region of Cnn-C inhibits binding to γ-TuRCs and therefore name this region the CM1 autoinhibition (CAI) domain. Removal of the CAI domain leads to robust binding, similar to that observed for the N-terminal region of Cnn-T. We identify two putative phosphorylation sites, one in the CAI domain ($T^{27}$) and one downstream of the CM1 domain ($S^{186}$), that promote binding to γ-TuRCs when phosphomimicked, suggesting that phosphorylation relieves CAI domain autoinhibition. We show that autoinhibition is important, as expressing a form of Cnn that binds to cytosolic γ-TuRCs leads to cytosolic microtubule nucleation and major defects during cell division. We further show that human CDK5RAP2 is inhibited from binding γ-TuRCs in the cytosol by a region downstream of the CM1 domain, showing that autoinhibition of binding is a conserved feature of CM1 domain proteins.

## Results

### The extreme N-terminal region of Cnn-C is inhibitory for γ-TuRC binding

We previously published evidence that different isoforms of Cnn bind γ-TuRCs with different affinities (Tovey et al., 2018). We found that bacterially purified maltose binding protein (MBP)–tagged N-terminal fragments of Cnn-T (MBP-Cnn-T-N) could coimmunoprecipitate more cytosolic γ-tubulin than the equivalent fragments of Cnn-C (MBP-Cnn-C-N). Both isoforms share a short sequence just proximal to the CM1 domain (residues 78–97 in Cnn-C), but differ in their extreme N-terminal region, which is 77 and 19 residues in Cnn-C and Cnn-T, respectively (Fig. 1 A). We had hypothesized that the larger extreme N-terminal region of Cnn-C may autoinhibit the CM1 domain, restricting its ability to bind γ-TuRCs. To address this directly and to confirm the in vitro results, we developed an in vivo assay with which γ-TuRC recruitment to different types of Cnn scaffolds formed within eggs could be monitored. To form scaffolds within eggs, we injected in vitro–generated mRNA encoding Cnn-C with phosphomimetic mutations within the PReM domain (Cnn-C-PReM$^{m}$; Fig. 1 B). mRNA is translated into protein within the egg and the phosphomimetic mutations cause the Cnn molecules to oligomerize into centrosome-like scaffolds throughout the cytosol (Conduit et al., 2014a; Fig. 1, C–F; and Fig. S1). To investigate how binding between Cnn and γ-TuRCs is regulated, we

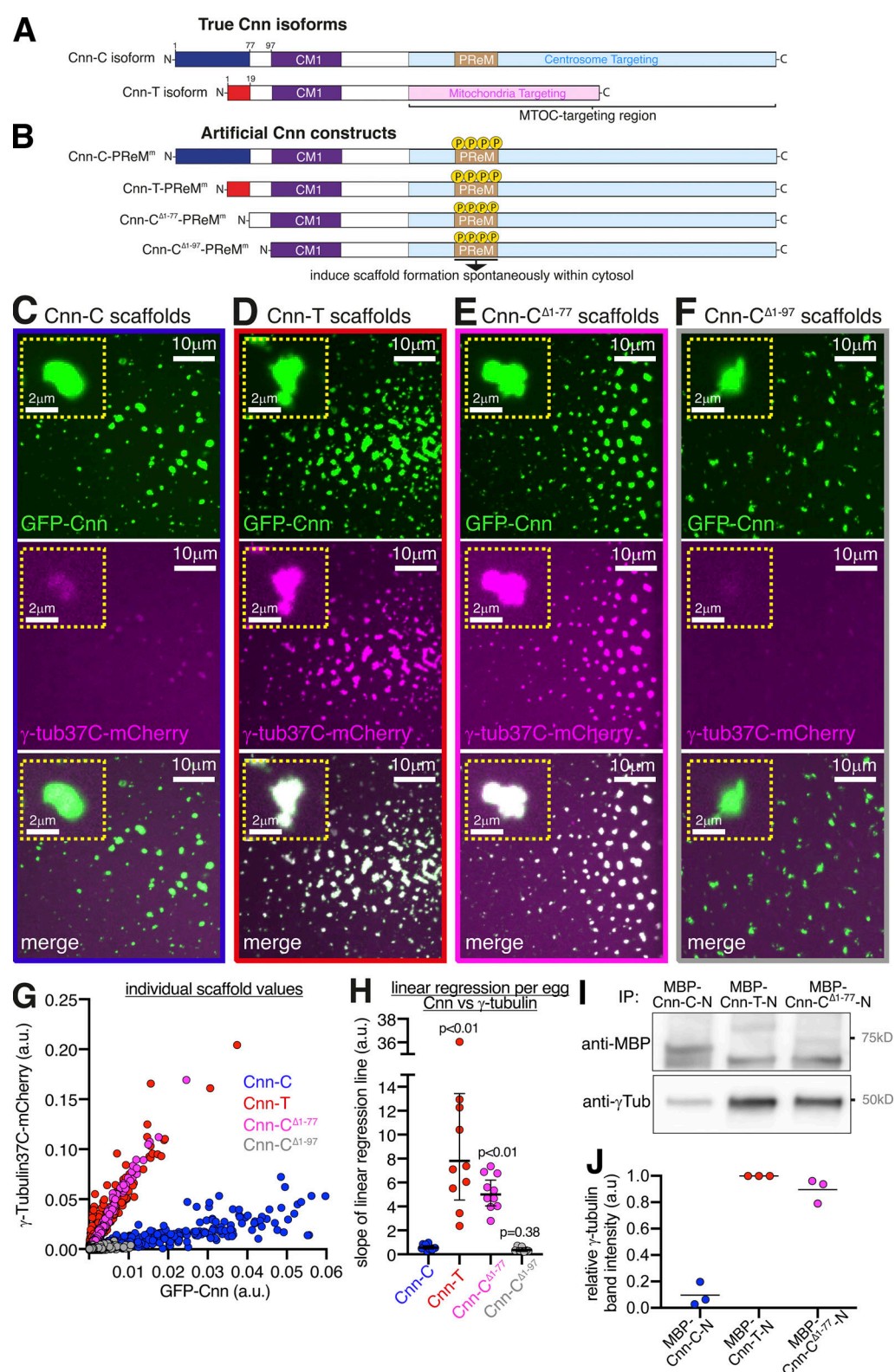

Figure 1. **The extreme N-terminal region of Cnn-C inhibits binding to γ-tubulin complexes. (A)** Diagram of the Cnn-C and testes-specific Cnn (Cnn-T) isoforms that exist in vivo. **(B)** Diagram of artificial Cnn proteins with differing N-terminal regions used to form Cnn scaffolds (induced by phosphomimetic mutations in the PReM domain; beige) via mRNA injection into unfertilized eggs. **(C–F)** Fluorescence images of unfertilized eggs expressing γ-tubulin37C-mCherry that were injected with mRNA encoding different types of artificial Cnn proteins, as indicated. Insets show representative examples of individual scaffolds. **(G)** Graph showing fluorescence intensity measurements (in arbitrary units) of γ-tubulin37C-mCherry and GFP-Cnn at Cnn-C ($n$ = 1,498 scaffolds; 12 eggs), Cnn-T ($n$ = 1,400 scaffolds; 10 eggs), Cnn-C$^{Δ1-77}$ ($n$ = 2,168 scaffolds; 10 eggs), or Cnn-C$^{Δ1-97}$ ($n$ = 400 scaffolds; 7 eggs) scaffolds. Each dot represents a single scaffold. **(H)** Graph shows slope values of linear regression lines calculated for scaffolds of different types. Each slope value represents an individual egg

that contained multiple scaffolds. The geometric mean and 95% CIs are indicated. P values are from comparisons to the Cnn-C mean using a one-way ANOVA of $log_{10}$ transformed data. **(I and J)** Western blot of a co-IP experiment (I) and quantification of γ-tubulin bands (J) showing the efficiency with which different MBP-tagged N-terminal fragments of Cnn, as indicated, coimmunoprecipitate γ-tubulin from embryo extracts. γ-Tubulin band intensities were normalized within each of three experimental repeats to the γ-tubulin band in the respective MBP-Cnn-T-N IP.

modified the N-terminal region of Cnn-C-PReM$^m$ (Fig. 1 B) and measured how efficiently fluorescently tagged γ-TuRC proteins could be recruited to the scaffolds.

We first compared the recruitment of endogenously tagged γ-tubulin37C-mCherry to GFP-tagged scaffolds formed from unmodified Cnn-C-PReM$^m$ with recruitment to scaffolds where the extreme N-terminal region (Fig. 1, A and B, dark blue) was either exchanged with the extreme N-terminal region of Cnn-T (Fig. 1, A and B, red; Cnn-T-PReM$^m$) or was removed (Cnn-C$^{\Delta 1-77}$-PReM$^m$). We also tested scaffolds in which all N-terminal amino acids up until the start of the CM1 domain were removed (Cnn-C$^{\Delta 1-97}$-PReM$^m$). For simplicity, we refer to these as Cnn-C, Cnn-T, Cnn-C$^{\Delta 1-77}$, and Cnn-C$^{\Delta 1-97}$ scaffolds, respectively, regardless of the fluorescent tag used. Initial observations suggested that γ-tubulin37C-mCherry associated much more readily with Cnn-T and Cnn-C$^{\Delta 1-77}$ scaffolds than with Cnn-C or Cnn-C$^{\Delta 1-97}$ scaffolds (Fig. 1, C–F). This was clear after plotting the GFP (Cnn) and mCherry (γ-tubulin37C) fluorescence values for individual scaffolds from multiple embryos per condition (Fig. 1 G). To quantify γ-tubulin37C recruitment, we performed linear regression for each egg separately and plotted the slope of these lines (S values, in arbitrary units). The mean S value provides an estimate for the relative binding affinity between the different forms of Cnn and γ-tubulin complexes (Fig. 1 H). The mean S values for Cnn-T scaffolds (7.81) and Cnn-C$^{\Delta 1-77}$ scaffolds (5.01) were ∼13-fold and ninefold higher, respectively, than the mean S value for Cnn-C scaffolds (0.57). Consistent with this, MBP-tagged N-terminal fragments of Cnn-T (MBP-Cnn-T-N) and Cnn-C$^{\Delta 1-77}$ (MBP-Cnn-C-N$^{\Delta 1-77}$) both coimmunoprecipitated more γ-tubulin from embryo extracts than N-terminal fragments of Cnn-C (MBP-Cnn-C-N; Fig. 1, I and J). Thus, the extreme N-terminal region of Cnn-C (Fig. 1, A and B, blue) is inhibitory for binding to γ-tubulin complexes.

The ability of Cnn-C$^{\Delta 1-77}$ to bind γ-tubulin complexes appeared to be dependent on the amino acids just upstream of the CM1 domain (aa 78–97), which are shared with Cnn-T (Fig. 1 A), as the mean S value for Cnn-C$^{\Delta 1-97}$ scaffolds (0.36) was not significantly different from that of Cnn-C scaffolds (0.57; Fig. 1 H). This is consistent with recent observations in *Saccharomyces cerevisiae*, showing that the equivalent amino acids within SPC110 make close contacts with SPC98$^{GCP3}$ (Brilot et al., 2021).

Cnn-T and Cnn-C$^{\Delta 1-77}$ scaffolds also recruited the γ-TuRC–specific component Grip75$^{GCP4}$-sfGFP better than Cnn-C scaffolds (Fig. 2, A–E). Similar to the recruitment of γ-tubulin37C, mean S values for Cnn-T (3.8) and Cnn-C$^{\Delta 1-77}$ (3.1) scaffolds were 10.3-fold and 8.4-fold higher, respectively, than the S value for Cnn-C (0.37) scaffolds (Fig. 2 E). (Note that these S values for Grip75$^{GCP4}$ cannot be compared directly to those obtained from analyzing γ-tubulin37C recruitment due to the different fluorescent tags used.) Moreover, a combination of Western blotting and mass spectrometry showed that bacterially purified MBP-Cnn-T-N

fragments could coimmunoprecipitate numerous other γ-TuRC components (Fig. S2).

The data collectively show that the extreme N-terminal region of Cnn-C (aa 1–77) inhibits binding to γ-TuRCs. We therefore name this region the CM1 autoinhibition or CAI domain.

## γ-TuRCs recruited by Cnn scaffolds appear to be able to generate dynamic microtubules

We next compared the ability of different scaffold types to organize microtubules. We imaged GFP-tagged Cnn-C (low γ-TuRC binding), Cnn-T, or Cnn-C$^{\Delta 1-77}$ (high γ-TuRC binding) scaffolds within eggs expressing the microtubule binding protein Jupiter-mCherry (Fig. 3, A–C) and performed a blind analysis to categorize eggs into those containing scaffolds that organized strong, weak, or no microtubule asters (Fig. 3 D). We also included a tubulin overlay category in which the Jupiter-mCherry signal did not extend beyond the GFP scaffold signal. Results show that Cnn-T and Cnn-C$^{\Delta 1-77}$ scaffolds were much more likely to organize microtubule asters than Cnn-C scaffolds (Fig. 3 D). This correlates with the increased recruitment of γ-TuRCs to Cnn-T and Cnn-C$^{\Delta 1-77}$ scaffolds (Fig. 1 H), suggesting that these γ-TuRCs are able to nucleate microtubules. While it is possible that some microtubules could have been generated independently of γ-TuRCs, a process that occurs by tubulin concentration at *C. elegans* SPD-5 condensates formed in vitro (Woodruff et al., 2017), the increased microtubule organizing capacity at Cnn-T and Cnn-C$^{\Delta 1-77}$ scaffolds (high γ-TuRC recruitment) compared with Cnn-C scaffolds (low γ-TuRC recruitment) suggests that γ-TuRC–mediated microtubule nucleation/organization is the predominant factor at these Cnn scaffolds.

Filming Cnn-T scaffolds through time revealed that the scaffolds could merge as well as be quite mobile, especially those that had microtubules emanating from just one side (Video 1). We could also observe events where spindle-like structures formed between adjacent Cnn-T or Cnn-C$^{\Delta 1-77}$ scaffolds (Fig. 3, E and F; and Videos 2 and 3), suggesting that the microtubules are dynamic and can be regulated by motor proteins. Giant Cnn-T scaffolds that rotated dragged their attached microtubules through the cytosol, indicating that the microtubules were robustly anchored to the scaffolds (Video 4). In summary, Cnn-T and Cnn-C$^{\Delta 1-77}$ scaffolds can recruit γ-TuRCs that are capable of nucleating and anchoring microtubules.

## Phosphomimetic mutations help relieve CAI domain–mediated autoinhibition

How could CAI domain–mediated autoinhibition be relieved to allow efficient binding to γ-TuRCs at centrosomes? Studies in human cells, *C. elegans*, and *S. cerevisiae* have shown that the binding of CM1 domain proteins to γ-TuSCs or γ-TuRCs is promoted by phosphorylation close to the CM1 domain (Hanafusa

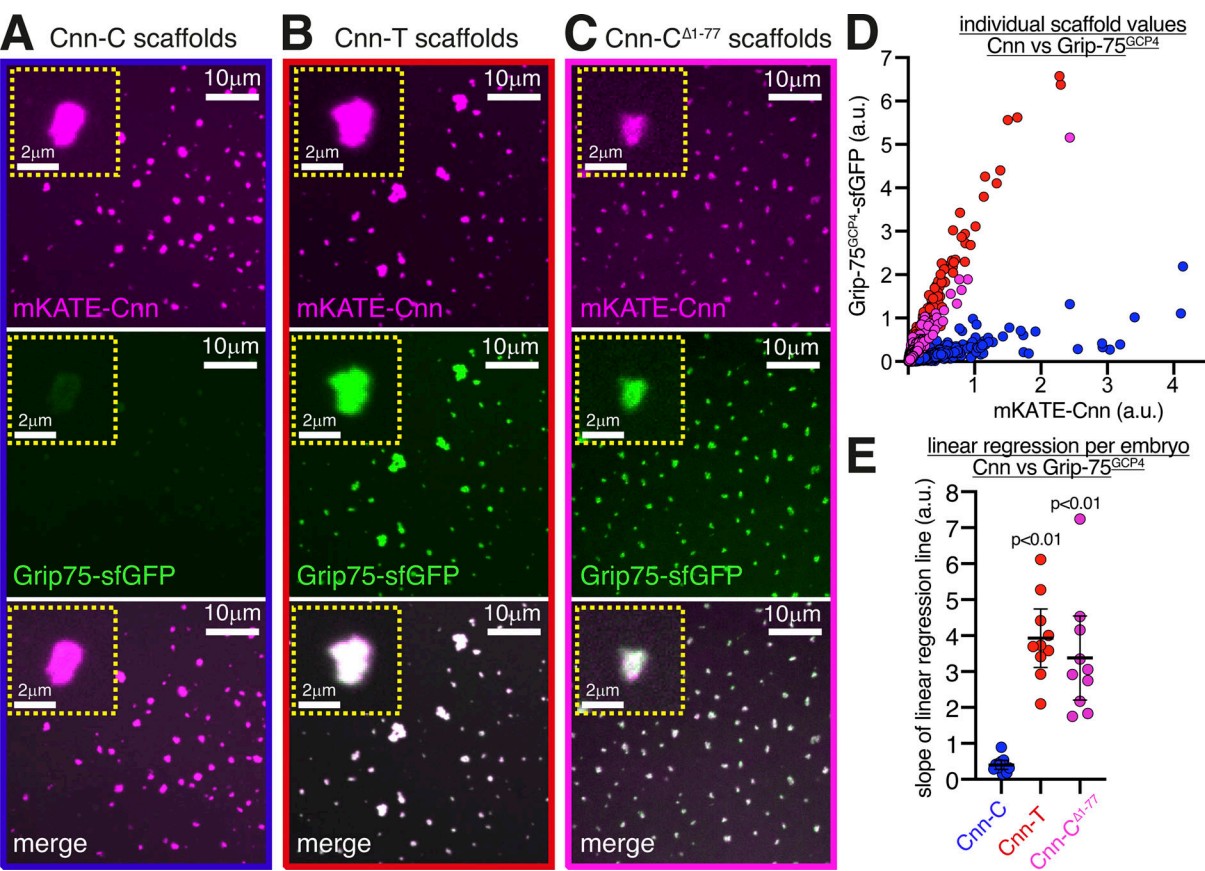

**Figure 2.** **The γ-TuRC–specific protein Grip75[GCP4] is recruited strongly to Cnn-T and Cnn-C[Δ1-77] scaffolds. (A–C)** Fluorescence images show mKATE-Cnn scaffolds of different types, as indicated, within eggs expressing endogenously tagged Grip75[GCP4]-sfGFP. Insets show representative examples of individual scaffolds. **(D)** Graph showing fluorescence intensity measurements (in arbitrary units) of Grip75[GCP4]-sfGFP and mKATE-Cnn at Cnn-C (*n* = 1,920 scaffolds; 12 eggs), Cnn-T (*n* = 1,650 scaffolds; 10 eggs), or Cnn-C[Δ1-77] (*n* = 2,599 scaffolds; 10 eggs). Each dot represents a single scaffold. **(E)** Graph shows slope values of linear regression lines calculated for scaffolds of different types. Each slope value represents an individual egg that contained multiple scaffolds. The mean and 95% CIs are indicated. P values are from comparisons to the Cnn-C mean using a one-way ANOVA.

et al., 2015; Ohta et al., 2021; Lin et al., 2014; Fig. S3 B). Moreover, Cnn-C binds γ-TuRCs and is phosphorylated only at centrosomes (Zhang and Megraw, 2007; Conduit et al., 2014a, 2014b), suggesting a possible link between binding and phosphorylation.

In an attempt to find phosphorylation sites that may relieve CAI domain inhibition, we aligned amino acids 1 to ~255 of Cnn-C homologues from various *Drosophila* species. We identified three putative phosphorylation patches (P1, P2, and P3) based on a high concentration of conserved serine and threonine residues (Fig. 4 A and Fig. S3). P1 represented the only region within the CAI domain with predicted secondary structure, corresponding to an α-helix (Fig. S3). We compared the amount of γ-tubulin that coimmunoprecipitated with purified MBP-tagged N-terminal fragments of Cnn-C containing phosphomimetic mutations (S>D or T>E) in all serine and threonine residues within either P1 (MBP-Cnn-C-N[P1]), P2 (MBP-Cnn-C-N[P2]), P3 (MBP-Cnn-C-N[P3]), or in all three patches (MBP-Cnn-C-N[P1-3]). The original MBP-Cnn-C-N (low binding) and MBP-Cnn-T-N (high binding) fragments were included as negative and positive controls, respectively. Of these phosphomimetic fragments, MBP-Cnn-C-N[P1] coimmunoprecipitated γ-tubulin most efficiently, although not as efficiently as

MBP-Cnn-T-N (Fig. 4, B and D). We therefore generated phosphomimetic fragments where either the proximal (S[21], S[22], T[27]) or distal (T[31], T[33], S[34]) three residues within P1 were mimicked (MBP-Cnn-C-N[P1a] or MBP-Cnn-C-N[P1b], respectively). We also phosphomimicked T[27] alone (MBP-Cnn-C-N[T27]), because T[27] is a putative Polo/Plk1 site and because a previous study reported centrosome defects when this site was mutated to alanine in vivo (Eisman et al., 2015). MBP-Cnn-C-N[P1a] and MBP-Cnn-C-N[T27], but not MBP-Cnn-C-N[P1b], coimmunoprecipitated more γ-tubulin than MBP-Cnn-C-N, although again not as much as MBP-Cnn-T-N (Fig. 4, C and D). In the scaffold assay, phosphomimicking T[27] also had a positive effect that was not as strong as that seen with Cnn-T or Cnn-C[Δ1-77] scaffolds. The mean S value for Cnn-C[T27] scaffolds (1.35) was ~2.4-fold higher than for Cnn-C scaffolds (0.57) but still lower than the S values for Cnn-T or Cnn-C[Δ1-77] scaffolds (Fig. 4 G). (Note that S values for Cnn-C[T27E] scaffolds and subsequent scaffolds analyzed below were compared with the S values for Cnn-C, Cnn-T, and Cnn-C[Δ1-77] scaffolds from Fig. 1 H.) Together, this suggested that while phosphorylation of T[27] may be involved in relieving CAI domain autoinhibition (or in directly increasing the binding affinity between Cnn-C and γ-TuRCs), it is not sufficient for robust γ-TuRC binding.

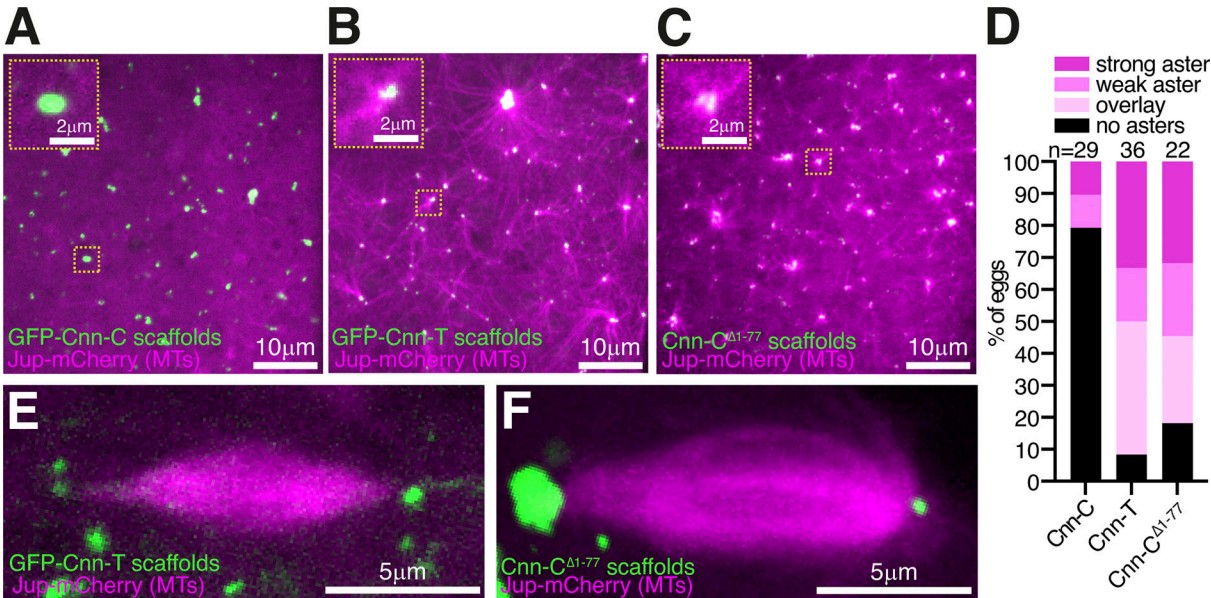

**Figure 3. Cnn-T and Cnn-C$^{\Delta 1-77}$ scaffolds organize microtubules more robustly than Cnn-C scaffolds. (A–C)** Fluorescence images of Cnn-C scaffolds (A), Cnn-T scaffolds (B), or Cnn-C$^{\Delta 1-77}$ scaffolds (C) within eggs expressing the microtubule marker Jupiter-mCherry. **(D)** Bar graph showing results of a blind categorization of eggs containing the different scaffold types based on the ability of the scaffolds within each egg to organize microtubule asters (numbers of eggs analyzed indicated above). **(E and F)** Fluorescence images showing that adjacent Cnn-T (E) or Cnn-C$^{\Delta 1-77}$ (F) scaffolds can organize spindle-like structures.

We therefore considered other putative phosphorylation sites. Phosphorylation slightly downstream of the CM1 domain promotes binding to γ-TuRCs in human and *C. elegans* CM1 domain proteins (Ohta et al., 2021; Hanafusa et al., 2015). While the sequence surrounding the CM1 domain is not conserved across diverse species (Fig. S3 B), we identified two serine residues (S173 and S186) downstream of the CM1 domain in Cnn that were conserved in *Drosophila* species (Fig. S3 A). These sites also mapped to a similar predicted coiled-coil region to the sites in human CDK5RAP2 and *C. elegans* SPD-5 (Fig. S3 B). While phosphomimicking S$^{173}$ had no effect, scaffolds with a phosphomimetic mutation at S$^{186}$ (Cnn-C$^{S186D}$ scaffolds) recruited ~3.8-fold more γ-tubulin than Cnn-C scaffolds (Fig. 4 G). Moreover, N-terminal fragments containing this mutation (Cnn-C-N$^{S186}$) coimmunoprecipitated γ-tubulin with a similar, if not higher, efficiency compared with the Cnn-T-N or Cnn-C$^{\Delta 1-77}$ fragments (Fig. 4 E). In addition, although not apparent in the scaffold assay (Fig. 4 G), phosphomimicking both T$^{27}$ and S$^{186}$ had a synergistic effect in the coimmunoprecipitation (co-IP) assay, where Cnn-C-N$^{T27E,S186D}$ fragments coimmunoprecipitated significantly more γ-tubulin than any other type of fragment (Fig. 4, E and F). The same pattern was seen when coimmunoprecipitating the γ-TuRC–specific protein Grip75$^{GCP4}$-sfGFP (Fig. 4 E). Unexpectedly, unlike in the co-IP assay, we did not see increased recruitment of Grip75$^{GCP4}$-sfGFP to scaffolds containing any of the N-terminal phosphomimetic mutations, including Cnn-C$^{T27E,S186D}$ scaffolds (Fig. 4 H). This suggested that these scaffolds recruit γ-TuSCs rather than γ-TuRCs, potentially explaining why they do not recruit γ-tubulin to the levels seen at Cnn-T or Cnn-C$^{\Delta 1-77}$ scaffolds (Fig. 4 G). Nevertheless, Cnn-C$^{T27E,S168D}$ scaffolds did organize microtubules more readily than Cnn-C scaffolds (Fig. 4 I; data compared with

that in Fig. 3 D), suggesting that the γ-tubulin complexes bound by the phosphomimetic forms of Cnn-C are at least semi-functional. Thus, while there are some differences between the scaffold assay and the co-IP assay, the data collectively suggest that phosphorylation at T$^{27}$ and, in particular, at S$^{186}$ helps to relieve CAI domain autoinhibition and promote the binding of Cnn-C to γ-TuRCs.

**Ubiquitous expression of Cnn-C containing the high binding-affinity Cnn-T N-terminal region has a dominant-negative effect and leads to fertility defects**

We next tested whether Cnn-C autoinhibition is important for cell and developmental fidelity in *Drosophila*. We generated a transgenic fly line by random insertion of a ubiquitously driven untagged Cnn-C construct in which its N-terminal region had been replaced with the N-terminal region of Cnn-T (pUbq-Cnn-C$^T$; Fig. 5 A). Based on our data so far, this form of Cnn should bind strongly to cytosolic γ-TuRCs but otherwise be regulated normally. We also generated a control line ubiquitously expressing untagged WT Cnn-C (pUbq-Cnn-C), whose binding to cytosolic γ-TuRCs should be restricted by the CAI domain.

It was difficult to generate a viable pUbq-Cnn-C$^T$ line and, once generated, it was difficult to maintain and combine with other alleles. Thus, all following experiments were performed with the pUbq constructs expressed in the presence of endogenous Cnn. By crossing pUbq-Cnn-C and pUbq-Cnn-C$^T$ females or males to WT flies and quantifying embryo hatching rates, we found that pUbq-Cnn-C$^T$ flies were less able to generate progeny than pUbq-Cnn-C flies, with males being more affected than females (Fig. 5 B). Western blots of embryo or testes extracts using different Cnn-C antibodies and a Cnn-T–specific antibody

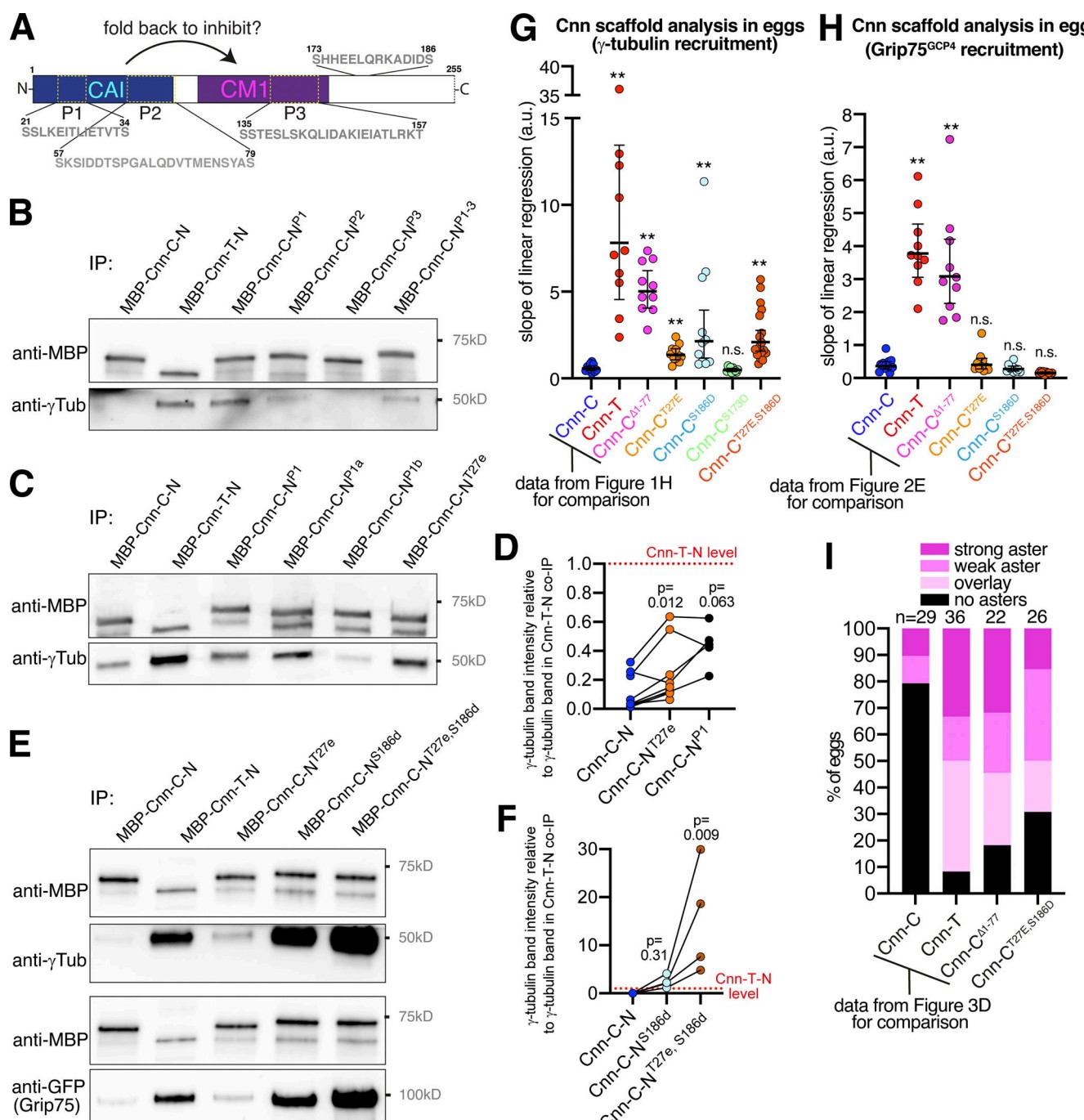

Figure 4. **Phosphomimetic mutations within the CAI domain and downstream of the CM1 domain promote binding to γ-tubulin complexes. (A)** A cartoon showing the N-terminal region (aa 1–255) of Cnn used in co-IP experiments. Regions of potential phosphorylation sites are indicated, with their amino acid sequence displayed. **(B–F)** Western blots of co-IP experiments (B,C, and E) and quantification of γ-tubulin bands (D and F) showing the efficiency with which different MBP-tagged N-terminal fragments of Cnn, as indicated, co-IP γ-tubulin from extracts of WT (B–F), or γ-tubulin (top panels in E) and Grip75[GCP4]-sfGFP (bottom panels in E) from extracts of Grip75[GCP4]-sfGFP–expressing embryos. In D and F, band intensities were normalized within each experiment to the γ-tubulin band in the respective MBP-Cnn-T-N IP. The connecting lines indicate data points obtained from within the same experiment. P values are from comparisons to the Cnn-C mean using either Wilcoxon matched-pairs signed rank tests (D; $n = 9$ for comparison with Cnn-C-N[T27]; $n = 5$ for comparison with Cnn-C-N[P1]) or a Dunn's multiple comparisons test (F; $n = 4$). **(G and H)** Graphs showing the S values from eggs expressing either γ-tubulin-mCherry (G) or Grip75[GCP4]-sfGFP (H) which contain the indicated scaffold types. Note that the data for Cnn-C, Cnn-T, and Cnn-C[Δ1-77] scaffolds is the same as in Fig. 1 H and Fig. 2 E to allow comparisons with the phosphomimetic scaffolds. In G: $n = 2,650$ scaffolds and 11 eggs for Cnn-C[T27] scaffolds, 1,803 scaffolds and 11 eggs for Cnn-C[T186] scaffolds, 2,482 scaffolds and 10 eggs for Cnn-C[T173] scaffolds, and 2,835 scaffolds and 18 eggs for Cnn-C[T27,S186] scaffolds. In H: $n = 1,448$ scaffolds and 10 eggs for Cnn-C[T27] scaffolds, 1,074 scaffolds and 10 eggs for Cnn-C[T186] scaffolds, and 943 scaffolds and 10 eggs for Cnn-C[T27,S186] scaffolds. The geometric mean and 95% CIs are indicated. **, $P < 0.01$. P values were from comparisons to the Cnn-C mean using a one-way ANOVA of $\log_{10}$ transformed data. **(I)** Bar graph showing the results of a blind categorization of eggs containing the different scaffold types based on the ability of the scaffolds within each egg to organize microtubule asters (numbers of eggs analyzed indicated above). Note that the data for Cnn-C, Cnn-T, and Cnn-C[Δ1-77] scaffolds is the same as in Fig. 3 D to allow comparisons with the phosphomimetic scaffolds.

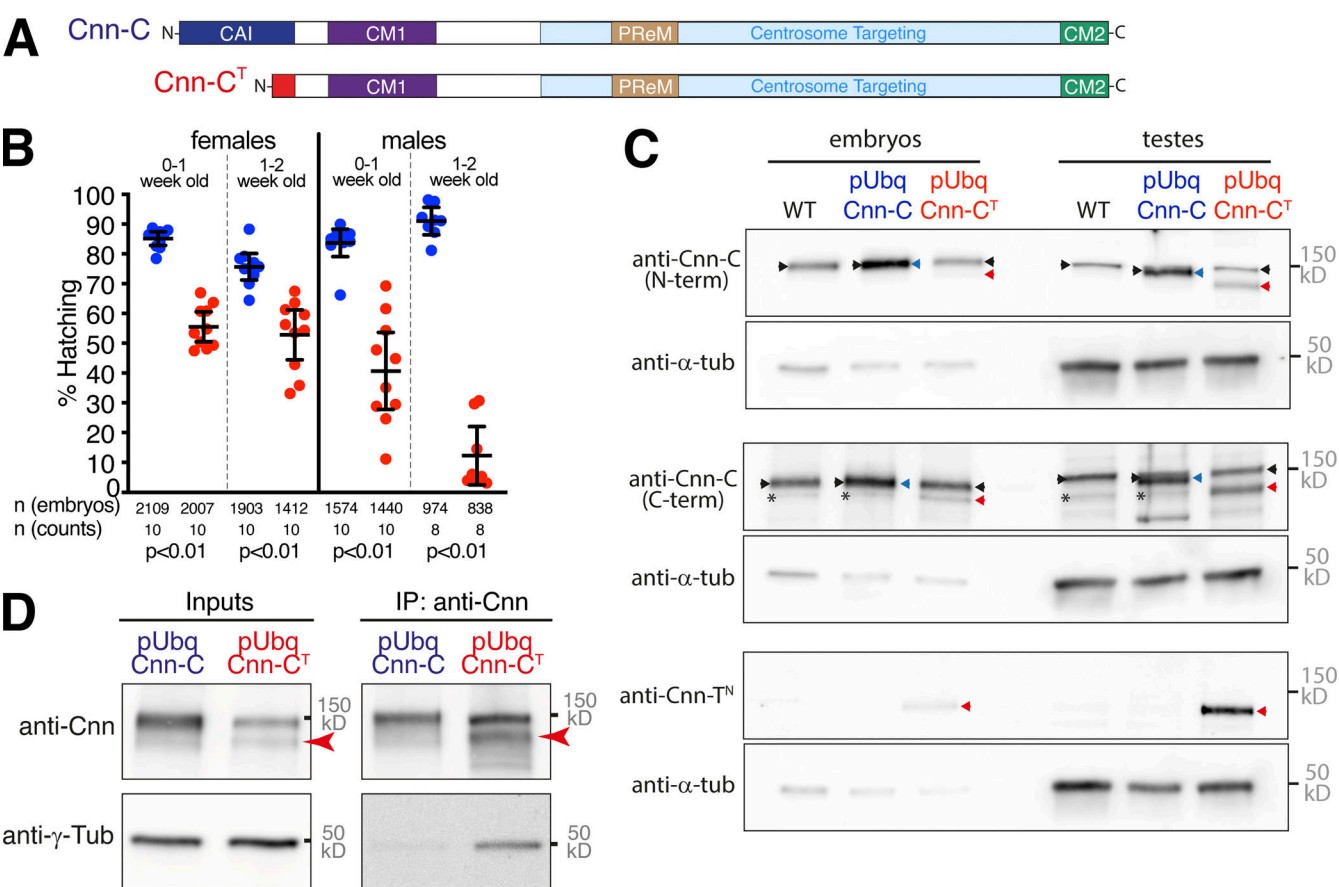

Figure 5. **Expression of pUbq-Cnn-C$^T$, which ectopically binds γ-TuRCs, reduces the ability of flies to generate progeny. (A)** Diagram of normal Cnn-C and chimeric Cnn-C$^T$ in which the CAI domain of Cnn-C (dark blue) is replaced by the shorter N terminus of Cnn-T (red). **(B)** Graph showing the proportion of embryos that hatched from crosses of WT flies to 0–1- or 1–2-wk-old pUbq-Cnn-C or pUbq-Cnn-C$^T$ males or females, as indicated. Means and 95% CIs are indicated. Total numbers of embryos counted and number of counts are indicated below. **(C)** Western blots of protein extracts from embryos and testes of WT, pUbq-Cnn-C, and pUbq-Cnn-C$^T$ flies, as indicated. Blots were probed with anti–γ-tubulin, anti–Cnn-C (N-term), anti–Cnn-C (C-term), and anti–Cnn-T$^N$ antibodies as indicated. Note that endogenous Cnn-C (black arrowheads) runs at the same height as pUbq-Cnn-C (blue arrowheads) on these blots, explaining the increased brightness of these bands in the pUbq-Cnn-C extract lanes. Note also that the C-terminal Cnn-C antibody recognizes an unspecific band (asterisks) of approximately the same size as pUbq-Cnn-C$^T$ (red arrowheads) and thus the pUbq-Cnn-C$^T$ band intensity would be lower in the absence of this unspecific band. **(D)** Western blot showing co-IP of γ-tubulin via anti-Cnn antibodies from embryo extracts expressing either pUbq-Cnn-C or pUbq-Cnn-C$^T$, as indicated. Red arrowhead indicates Cnn-C$^T$. Note that, given the low expression of pUbq-Cnn-C$^T$ within embryos, gel loading of the IP lanes was adjusted to better balance the amount of Cnn protein per lane.

showed that the level of pUbq-Cnn-C$^T$ (red arrowheads) relative to endogenous Cnn-C (black arrowheads) was higher in testes extracts compared with embryo extracts (Fig. 5 C). In the embryo extracts, the pUbq-Cnn-C$^T$ band was much weaker than the endogenous Cnn-C band, which is unusual for pUbq-driven Cnn constructs (unpublished data), suggesting its expression was being suppressed. In contrast, the pUbq-Cnn-C$^T$ band was of a similar intensity to, if not higher than, the endogenous Cnn-C band in the testes extracts. We therefore conclude that, relative to endogenous Cnn-C, pUbq-Cnn-C$^T$ is weakly expressed within the maternal germline but is expressed to levels similar to endogenous Cnn within the testes. While other factors could be involved, such as cell-specific effects of Cnn to γ-TuRC binding, these differences in the expression levels of pUbq-Cnn-C$^T$ between cells could explain the difference in the ability of male and female flies to generate progeny.

**Misregulation of binding to γ-tubulin complexes results in ectopic microtubule nucleation and defects during cell division**

The failure of pUbq-Cnn-C$^T$ flies to generate normal numbers of progeny suggested that ectopic binding of Cnn to γ-TuRCs leads to cellular defects during germline or early development. co-IPs from embryo extracts confirmed that pUbq-Cnn-C$^T$ binds γ-TuRCs more efficiently than pUbq-Cnn-C (Fig. 5 D). We immunostained female and male germline tissues to investigate any potential defects within their cells. There were no obvious defects within oocytes from either pUbq-Cnn-C or pUbq-Cnn-C$^T$ females with regard to polarised microtubule-based transport, as the position of nuclei and Gurken and Staufen proteins were normal (Fig. S4, A–C). We did, however, frequently observe defects in fixed and stained syncytial embryos from pUbq-Cnn-C$^T$ females (Fig. 6, D–F) compared with embryos from pUbq-Cnn-C females (Fig. 6, A–C). These defects included an apparent excess of cytosolic microtubules, unusually

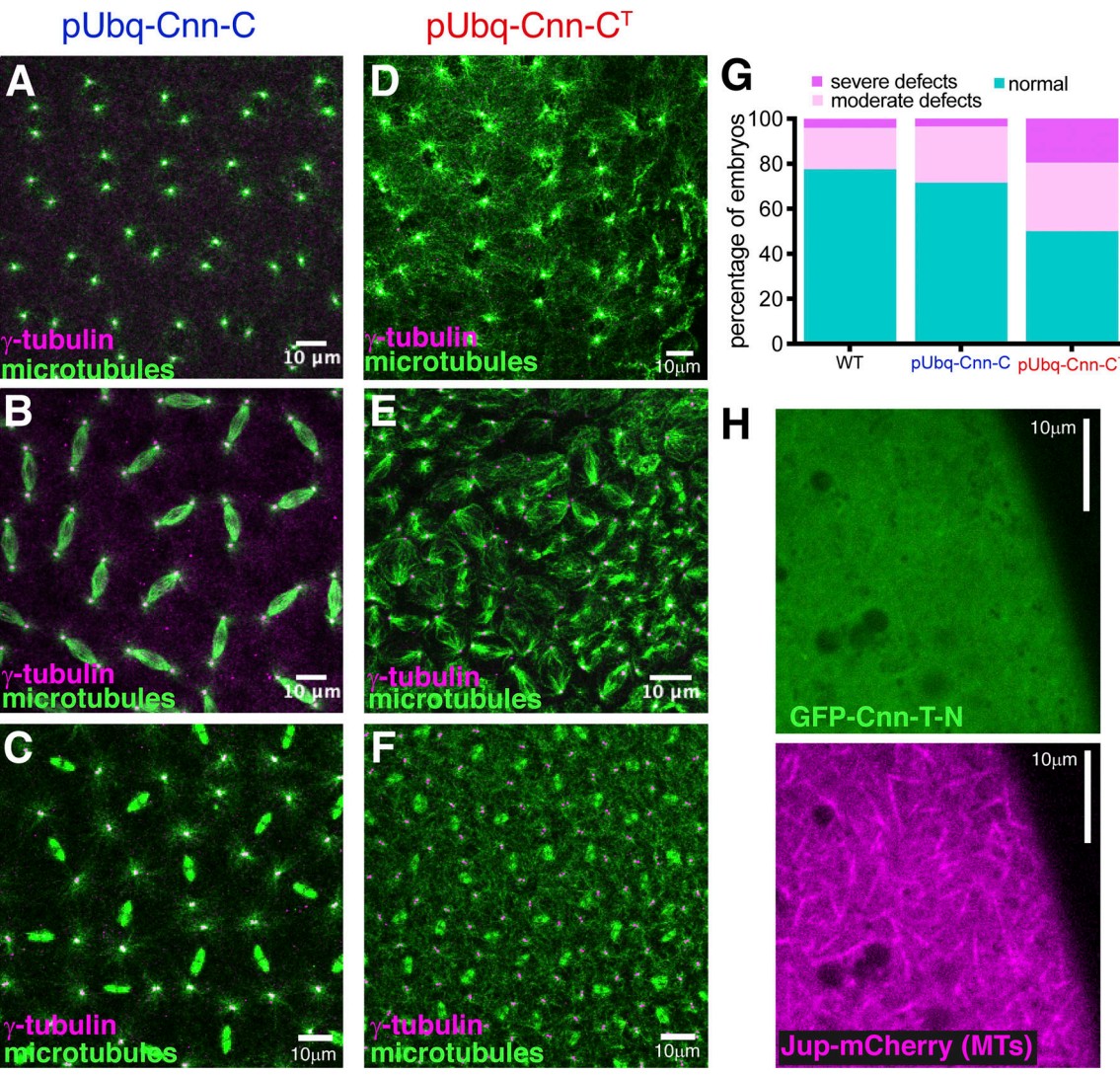

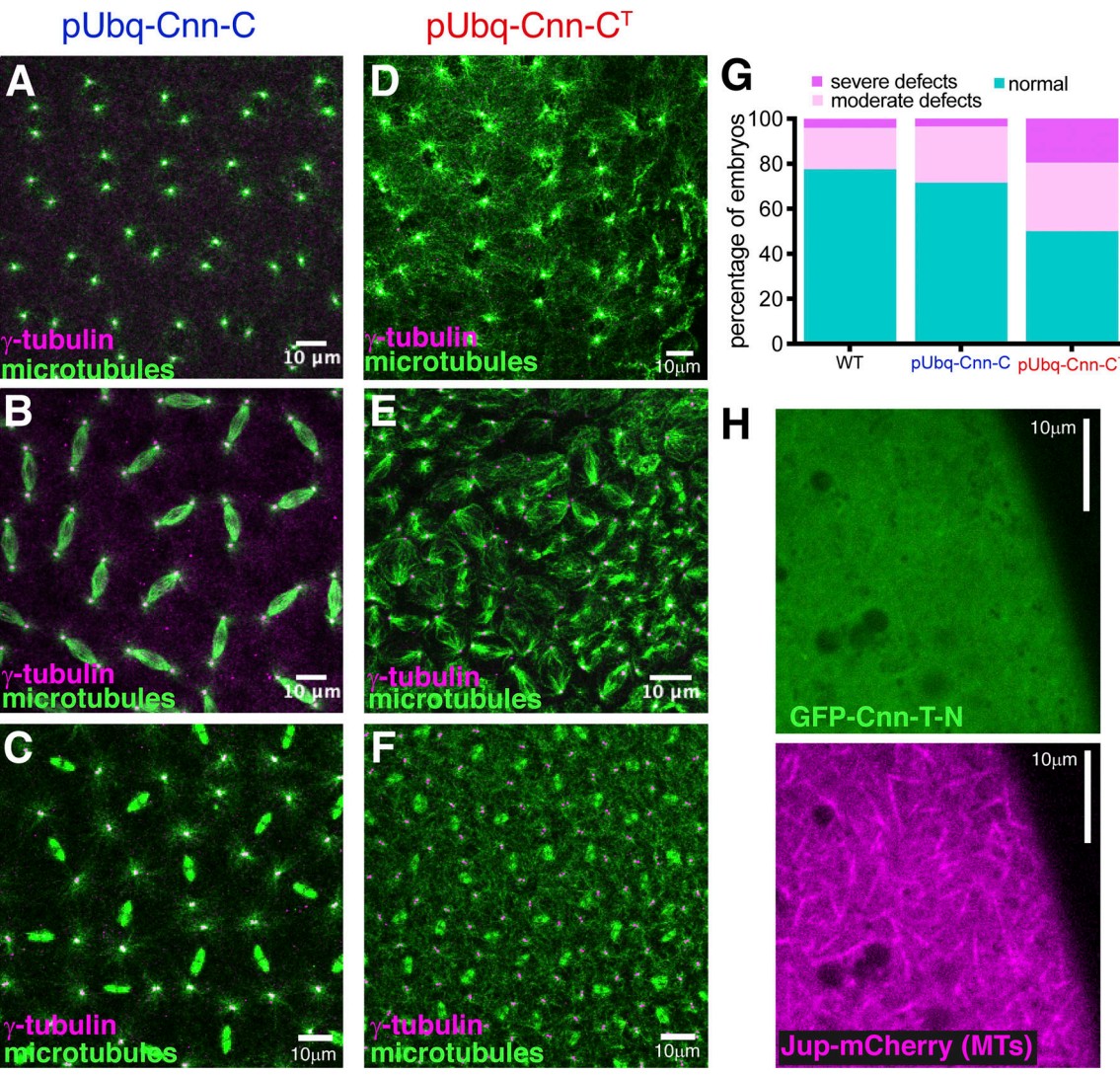

**Figure 6.** **Expression of pUbq-Cnn-C[T] increases the frequency of nuclear and spindle defects observed within syncytial embryos. (A–F)** Fluorescence images of syncytial embryos expressing either pUbq-Cnn-C (A–C) or pUbq-Cnn-C[T] (D–F) in either S-phase/prophase (A and D) Metaphase (B and E), or telophase (C and F). Note the apparent high density of cytosolic microtubules that can be (but are not always) observed in pUbq-Cnn-C[T] embryos, along with major organization defects. **(G)** Graph showing results from a blind categorization of WT ($n = 49$), pUbq-Cnn-C ($n = 88$), or pUbq-Cnn-C[T] ($n = 36$) embryos based on the presence or absence of moderate or severe nuclear or spindle defects. **(H)** Fluorescence images of an egg expressing the microtubule marker Jupiter-mCherry (magenta) that had been injected with mRNA encoding GFP-Cnn-T-N (green).

bright microtubule asters, and nuclear organization defects during S-phase (Fig. 6 D); highly disorganized spindles during M-phase (Fig. 6 E); and an apparent excess of cytosolic microtubules during telophase (Fig. 6 F). In a blind analysis of embryos, severe and moderate defects were observed in a higher proportion of embryos from pUbq-Cnn-C[T] females (19.4% severe and 30.6% moderate) than from WT (4.1% and 18.4%, respectively) or pUbq-Cnn-C (3.4% and 25%, respectively) females (Fig. 6 G). Broadly, the categorization of embryo defects in Fig. 6 (G) reflects the observed hatching rates in Fig. 5 (B), assuming that embryos with moderate and severe defects often fail in development. While half of the embryos from pUbq-Cnn-C[T] females were normal, this could reflect the relatively low expression of pUbq-Cnn-C[T] in the female germline (Fig. 5 C).

To directly test whether binding of Cnn to cytosolic γ-TuRCs could promote ectopic microtubule nucleation within embryos, we injected unfertilized eggs with mRNA encoding GFP-Cnn-T-N, which efficiently binds cytosolic γ-TuRCs. We found that 9 of 12 of these eggs displayed dynamic microtubules throughout their cytosol (Fig. 6 H and Video 5). This was not observed in any of the 27 control-injected eggs. This effect is similar to that observed when expressing CM1 domain fragments within human and fission yeast cells (Choi et al., 2010; Cota et al., 2017; Hanafusa et al., 2015; Lynch et al., 2014) and suggests that CM1 domain binding to γ-TuRCs also promotes microtubule nucleation in *Drosophila*.

Consistent with a very strong reduction in the ability of pUbq-Cnn-C[T] males to generate progeny, defects were frequently observed within their testes, where production of sperm

involves a series of mitotic and meiotic cell divisions. When meiosis progresses normally, the 64 round spermatids cells within the resulting cyst all contain a similarly sized phase-light nucleus and phase-dark nebenkern, which is an accumulation of mitochondria that were segregated during meiosis. This was true in round spermatids from pUbq-Cnn-C testes (Fig. 7, A and C), but not in round spermatids from pUbq-Cnn-C$^T$ testes (Fig. 7, B and C), suggesting that pUbq-Cnn-C$^T$ expression results in problems in chromosome segregation and cytokinesis. Indeed, a high density of cytosolic microtubules and clear meiotic defects were observed in spermatocytes within fixed and stained pUbq-Cnn-C$^T$ testes, but not pUbq-Cnn-C testes. Defects were observed at various developmental stages and included cells with incorrect numbers of nuclei and centrosomes as well as cells containing multiple spindles (Fig. 7, D and E; and Fig. S5, A and B). Thus, it appears that ectopic binding of pUbq-Cnn-C$^T$ to γ-TuRCs within spermatocytes leads to excessive cytosolic microtubules and major defects during meiosis.

### Human CDK5RAP2 binding to γ-TuRCs is also regulated by autoinhibition, but the precise mechanism differs from *Drosophila*

To examine whether autoinhibition is a conserved feature of CM1 domain proteins, we tested the ability of various N-terminal fragments of human CDK5RAP2 (Fig. 8 A) to coimmunoprecipitate γ-tubulin from HEK cell extracts. The reported CM1 domain spans aa 58–126 of CDK5RAP2 (Sawin et al., 2004; Zhang and Megraw, 2007; Fig. 8 A) and a fragment spanning aa 1–210 was less efficient at coimmunoprecipitating γ-tubulin than a fragment spanning aa 51–100 (also known as γ-TuNA; Choi et al., 2010; Fig. 8, B and C). This indicated that sequences either upstream or downstream of γ-TuNA are inhibitory for binding to γ-TuRCs. A fragment that included the sequence upstream of γ-TuNA (aa 1–100) coimmunoprecipitated γ-tubulin more efficiently than γ-TuNA (Fig. 8 B), suggesting that, unlike in *Drosophila* Cnn, the sequence upstream of the CM1 domain is not inhibitory but is instead required for efficient binding. In contrast, a fragment that included the sequence downstream of γ-TuNA (aa 51–210) was less efficient than γ-TuNA at coimmunoprecipitating γ-tubulin (Fig. 8 B). This suggests that the sequence downstream of the CM1 domain in CDK5RAP2 inhibits binding to γ-TuRCs. Thus, while autoinhibition appears to regulate the binding of CDK5RAP2 to γ-TuRCs as in flies, the precise mechanism seems to vary between species.

### Discussion

We have shown that the extreme N-terminal region of Cnn-C, which we named the CAI domain, inhibits Cnn-C from binding to γ-TuRCs. This autoinhibition is important because expressing a form of Cnn that readily binds γ-TuRCs within the cytosol leads to spindle and cell division defects, possibly via the ectopic activation of γ-TuRCs. Phosphomimicking experiments suggest that phosphorylation at sites close to the CM1 domain relieves autoinhibition of Cnn-C and promotes binding to γ-TuRCs. This is consistent with Cnn-C being phosphorylated specifically at

centrosomes during mitosis (Conduit et al., 2014a) where binding and activation of γ-TuRCs is believed to take place. In addition, human CDK5RAP2 is inhibited from binding cytosolic γ-TuRCs by the region downstream of the CM1 domain. Thus, while the precise mechanism may vary, it appears that autoinhibition is a conserved feature of CM1 domain proteins.

There is considerable evidence, including the work presented here, showing that binding of CM1 domain proteins to γ-tubulin complexes stimulates microtubule nucleation (Choi et al., 2010; Muroyama et al., 2016; Hanafusa et al., 2015; Cota et al., 2017; Lynch et al., 2014), but the reason remains unclear. One possibility is that binding leads to conformational changes in γ-TuRCs, but human and *Xenopus* γ-TuRCs bound by CM1 domain fragments remain in an open, seemingly inactive, conformation (Wieczorek et al., 2020; Liu et al., 2020). Whether this is due to a low stoichiometry of binding remains unclear, but binding of the CM1 domain to *S. cerevisiae* γ-TuSCs/γ-TuRCs does result in structural changes that possibly promote nucleation activity (Brilot et al., 2021). It is also possible that CM1 domain binding has a context-specific effect. Adding CM1 domain fragments to purified γ-TuRCs within *Xenopus* egg extracts had a greater effect on nucleation efficiency when the extract was supplemented with activated Ran (Liu et al., 2020), and we find that expression of pUbq-Cnn-C$^T$ leads to defects within specific cell types—although these differences could simply be due to the differences in expression levels. Clearly, we need a better understanding of exactly how CM1 domain binding promotes microtubule nucleation.

Phosphorylation seems to be an important mechanism for promoting binding between CM1 domain proteins and γ-TuRCs. This is true for human CDK5RAP2, *C. elegans* SPD-5, and *S. cerevisiae* SPC110, where the phosphorylation sites that promote binding have been identified either upstream or downstream of the CM1 domain (Hanafusa et al., 2015; Lin et al., 2014; Ohta et al., 2021). We show that phosphomimicking sites that are both upstream and downstream of the CM1 domain also promotes binding of *Drosophila* Cnn-C to γ-TuRCs. We predict that phosphorylation helps to relieve the autoinhibition imposed by the CAI domain and directly increases binding affinity between Cnn-C and γ-TuRCs. We find that phosphomimicking S$^{186}$ alone allows robust binding to γ-TuRCs, suggesting that phosphorylating this single site is sufficient for full relief of autoinhibition, at least in vitro. Phosphomimicking T$^{27}$ alone has a more subtle effect but has a strong effect when combined with the S$^{186}$ phosphomimic mutation. This suggests that phosphomimicking T$^{27}$ increases the binding affinity between Cnn-C and γ-TuRCs, rather than relieving autoinhibition, and thus has a minimal effect when Cnn-C is autoinhibited (when S$^{186}$ is not phosphomimicked), but has a strong effect when Cnn-C inhibition is relieved (by S$^{186}$ phosphomimicking). This would suggest that the CAI domain, which contains T$^{27}$, is also involved in binding to γ-TuRCs once inhibition is relieved. A role for the region upstream of the CM1 domain in binding to γ-TuRCs may be conserved, as our data show that this region promotes binding of human CDK5RAP2 to γ-TuRCs.

In future, it will be important to understand how the CAI domain inhibits the CM1 domain. We previously postulated that

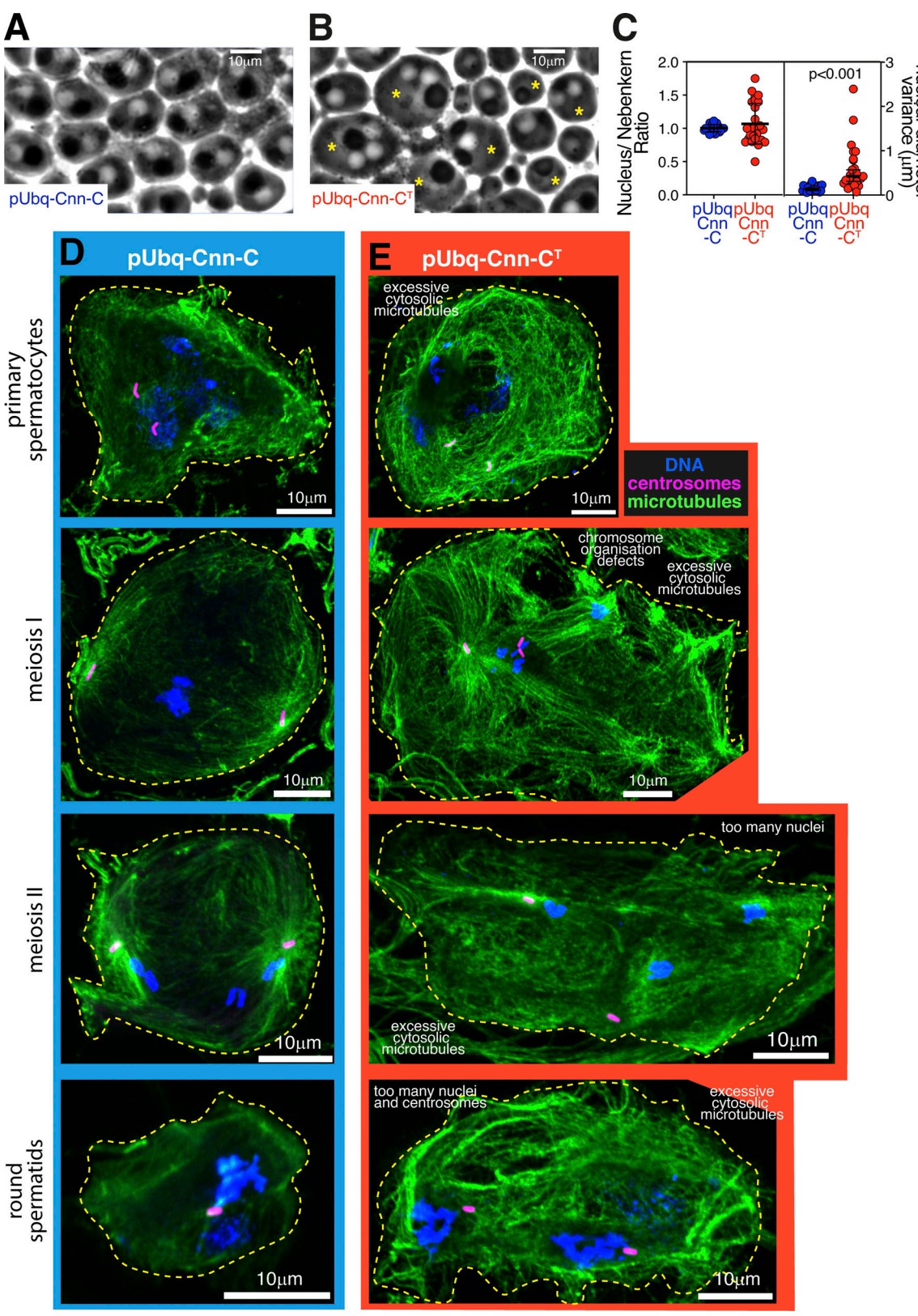

Figure 7. **Expression of pUbq-Cnn-C^T results in major defects during male meiosis. (A and B)** Phase-contrast images showing round spermatids from testes of flies expressing pUbq-Cnn-C (A) or pUbq-Cnn-C^T (B). Alterations in nucleus: nebenkern ratio (normally 1:1, asterisks in right panel) and size (normally approximately equal) indicate defects in cytokinesis and karyokinesis. **(C)** Graph showing quantification of the nucleus:nebenkern ratio (left panel: means and SDs indicated) and variance in nuclear diameter (right panel: geometric means and 95% CIs indicated, P value from an unpaired $t$ test of $\log_{10}$-tranformed data) in pUbq-Cnn-C ($n$ = 22 cysts) and pUbq-Cnn-C^T ($n$ = 27 cysts) testes. **(D and E)** Fluorescence images showing spermatocytes or round spermatids at different developmental stages, as indicated, from testes of flies expressing pUbq-Cnn-C (D) or pUbq-Cnn-C^T (E) stained for microtubules (green, α-tubulin), centrosomes (pink, asterless), and DNA (blue). Defects within cells expressing pUbq-Cnn-C^T include an apparent high density of cytosolic microtubules, abnormal spindles, and too many nuclei.

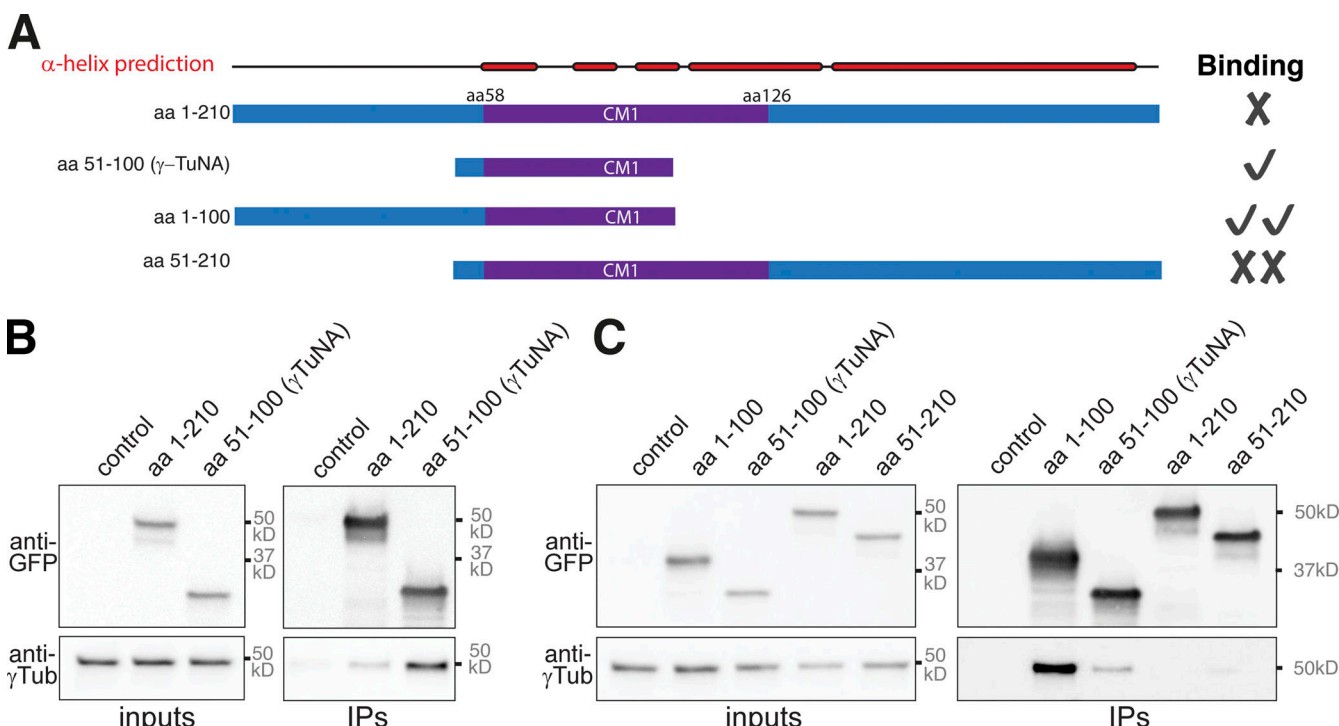

Figure 8. **The region downstream of the CM1 domain in human CDK5RAP2 is inhibitory for binding to γ-TuRCs. (A)** Cartoon depicting the various CDK5RAP2 N-terminal fragments used in IP experiments and indicating their relative γ-TuRC binding affinity. **(B and C)** Western blots of co-IP experiments from HEK cell extracts probed for the various GFP-tagged CDK5RAP2 fragments (top) and γ-tubulin (bottom).

it might fold back and sterically inhibit the CM1 domain (Tovey et al., 2018). Our data are consistent with this possibility and, in our view, this is the most likely explanation. A similar mechanism has also been proposed in *C. elegans* (Ohta et al., 2021). Nevertheless, there are alternative possibilities, including that the CAI domain could recruit other proteins that interfere with CM1 domain binding. In any case, it will be interesting to compare how autoinhibition is achieved in different homologues, especially given that the region downstream, not upstream, of the CM1 domain appears to mediate inhibition in human CDK5RAP2.

Importantly, our data also highlight differences in how binding between CM1 domain proteins and γ-TuRCs is regulated within different cell types and at different MTOCs. Testes-specific Cnn-T isoforms lack the CAI domain and recruit γ-TuRCs to mitochondria in developing sperm cells (Chen et al., 2017). While we cannot rule out that Cnn-T contains autoinhibitory domains in its C-terminal region, we have shown that the N-terminal region of Cnn-T can bind efficiently to γ-TuRCs in the apparent absence of any upstream regulatory events. The surface of mitochondria may well lack the kinases that regulate Cnn-C at centrosomes and it would therefore seem appropriate that Cnn-T can bind γ-TuRCs in the absence of phosphorylation. This would make most sense should Cnn-T isoforms be predominantly expressed postmeiotically, which is common for testes-specific genes (White-Cooper, 2012). This is because the binding and activation of cytosolic γ-TuRCs by Cnn-T isoforms may perturb dividing cells but presumably would not perturb developing

sperm cells, which have a shrinking cytosol and no need to form a spindle.

In summary, the data presented here provide important insights into how and why binding of CM1 domain proteins to γ-TuRCs is regulated. Future studies will help elucidate the precise mechanism underlying autoinhibition of the CM1 domain and how this may vary between species. It will also be important to determine whether CM1 domain binding directly activates γ-TuRCs and, if not, how CM1 domain binding promotes microtubule nucleation.

## Materials and methods
### DNA cloning
5-α competent *Escherichia coli* cells (high efficiency; New England Biolabs) were used for bacterial transformations. DNA fragments were purified using QIAquick Gel Extraction kits (Qiagen), and plasmid purification was performed using QIAprep Spin Miniprep kits (Qiagen). Phusion high-fidelity PCR master mix with HF buffer (Thermo Fisher Scientific) was used for PCRs.

### Transgenic *Drosophila* lines
All endogenously tagged lines were made using CRISPR combined with homologous recombination by combining the presence of a homology-repair vector containing the desired insert with the appropriate guide RNAs and Cas9. The γ-tubulin37C-mCherry and Grip128-sfGFP alleles were generated by inDroso. For γ-tubulin37C-mCherry, eggs from nos-Cas9–expressing

females were coinjected with a plasmid encoding the expression of dual guides targeting each side of the 3′ untranslated region, 5′-TACACATATCAAGATACATG-3′ and 5′-CCCAGATCG ATTATCCCCAG-3′, and a plasmid containing an SSSS-mCherry-3′ untranslated region-LoxP-3xP3-dsRED-Lox P cassette flanked by homology arms (the multiserine insert acts as a flexible linker). After screening for dsRED, the selection marker was excised by Cre recombination. For Grip128-sfGFP, eggs from nos-Cas9–expressing females were coinjected with a plasmid encoding the expression of a single guide containing the target sequence 5′-ATGGGGCACACT GGAGTTGA-3′ and with a pBluescript plasmid containing sfGFP and linker sequence (4× GlyGlySer) flanked on either side by 1.5 kb of DNA homologous to the genomic locus surrounding the 3′ end of the appropriate coding region. The homology vector was made within the laboratory (and sent to InDroso) by HiFi assembly (New England Biolabs) of PCR fragments generated from genomic DNA prepared from nos-Cas9 flies (using MicroLYSIS, Microzone) and a vector containing the sfGFP tag (Drosophila Genome Resource Centre, 1314). Screening for the insert was performed with the following primers: 5′-AGGAAGATGCGAACACACGT-3′ and 5′-GTACAG CTCATCCATGCCCA-3′.

The Grip75-sfGFP and Grip163-sfGFP lines were made within the laboratory following a similar approach to that used previously (Tovey et al., 2018; Mukherjee et al., 2020). Flies expressing a single guide RNA containing the target sequence 5′-CAAAAACATCGTATTCATG-3′ or 5′-ACCACTATTACAAGG TATCT-3′ for Grip75-sfGFP or Grip163-sfGFP, respectively, were crossed to nos-Cas9–expressing females and the resulting embryos were injected with homology vectors by the Department of Genetics Fly Facility. The homology vectors comprised a pBluescript plasmid containing sfGFP and linker sequence (4×X GlyGlySer) flanked on either side by 1.5 kb of DNA homologous to the genomic locus surrounding the 3′ end of the appropriate coding region. The homology vectors were made as for Grip128-sfGFP. F1 and F2 males were screened by PCR using the following primers: Grip75-sfGFP: 5′-GAGAAGTTTGCGCATATGACCC-3′ and 5′-AGCAGCACCATGTGATCGCGC-3′; and Grip163-sfGFP: 5′-AGTCGCAGTCCTTTATTGTGG-3′ and 5′-AGCAGCACCATG TGATCGCGC-3′.

pUbq-Cnn-C and pUbq-Cnn-C^T were made from a pDONR-Cnn-C vector (gift from Jordan Raff, Sir William Dunn School of Pathology, University of Oxford, Oxford, UK). To generate a Cnn-T–specific N-terminal region of Cnn, an appropriate DNA fragment (made by GENEWIZ, based on the FlyBase sequence of Cnn-T) was synthesized and amplified by PCR and used to replace the N-terminal region of Cnn in a pDONR-Cnn-C vector cut with XmaI. The pDONR-Cnn-C and newly made pDONR-Cnn-T vectors were then inserted into a pUbq transformation vector (gift from Jordan Raff) by Gateway cloning (Thermo Fisher Scientific). All DNA vectors were injected into embryos by the Department of Genetics Fly Facility.

The Jupiter-mCherry line used to monitor microtubule nucleation was a gift from Jordan Raff's laboratory. The original line was a GFP trap line from Daniel St. Johnston's laboratory and the GFP was replaced with mCherry.

## Recombinant protein cloning, expression, and purification

Fragments of Cnn-C-N and Cnn-T-N used in co-IP experiments were amplified from the pDONR-Cnn-C and pDONR-Cnn-T vectors described above by PCR and inserted into a pDEST-HisMBP (#11085; Addgene) vector by Gateway cloning (Thermo Fisher Scientific). Proteins were expressed in *E. coli* (BL21-DE3) and purified using affinity chromatography. MBP-tagged fragments were purified by gravity flow through amylose resin (New England Biolabs) and step elution in maltose. The concentration of each fraction was determined on a Nanodrop and peak fractions were diluted 1:1 with glycerol and stored at –20°C. Truncated fragments of Cnn-C were made by modification of the pDONR-Cnn-C-N entry clone. The N-terminal region was removed by a Quikchange reaction (Agilent Technologies), and the resulting shortened fragment was inserted into the pDEST-HisMBP destination vector via a Gateway reaction.

Phosphomimetic fragments were created by modifying the pDONR-Cnn-C-N entry clone. The pDONR-Cnn-C-N backbone was linearized by PCR or by digestion, omitting the phospho-patch to be replaced. Phosphomimetic patches in which all S/T residues were swapped for D/E residues, respectively, were synthesized either by PCR using two overlapping primers or by GENEWIZ. They were inserted into the linear backbone by HiFi assembly (New England Biolabs). Entry clones were checked by restriction enzyme digest and sequencing before being inserted into the pDEST-HisMBP destination vector via a Gateway reaction.

pRNA vectors were made by modification of the pDONR-Cnn-C-PReM^P vector containing phosphomimetic mutations in the PReM domain (Conduit et al., 2014a). N-terminal variants were introduced by restriction digests (SspI-HF and AatII) of pDONR-Cnn-C, pDONR-Cnn-T, and pDONR-Cnn-C-PReM^P entry clones. Fragments were combined as necessary by HiFi assembly to create new pDONR vectors that were inserted into a pRNA-GFP or pRNA-mKate destination vector (Conduit et al., 2014a) via a Gateway reaction. The Cnn-T-N fragment was inserted directly into pRNA-GFP destination vectors via Gateway cloning.

Fragments of CDK5RAP2 were synthesized by GENEWIZ, amplified by PCR, and cloned into a pCMV-GFP vector (gift from Jens Lüders, Institute for Research in Biomedicine [IRB Barcelona], The Barcelona Institute of Science and Technology, Barcelona, Spain) by restriction digest and HiFi assembly (New England Biolabs).

Primers used are listed in Table 1.

## Immunoprecipitation

1 g/ml of embryos were homogenized with a hand pestle in homogenization buffer containing 50 mM Hepes (pH 7.6), 1 mM MgCl$_2$, 1 mM EGTA, 50 mM KCl supplemented with PMSF 1:100, Protease Inhibitor Cocktail (1:100; Sigma-Aldrich), and DTT (1 M, 1:1,000). Extracts were clarified by centrifugation twice for 15 min at 16,000 rcf at 4°C.

For the MBP-Cnn fragment immunoprecipitates, 30 µl magnetic ProteinA dynabeads (Life Technologies) coupled to anti-MBP antibodies (gift from Jordan Raff) were incubated with an excess of purified MBP-Cnn fragments and rotated for 1 h at

Table 1. **Primers used in the study**

| | Forward primer | Reverse primer |
|---|---|---|
| Cnn-C-N fragment | 5'-GGGGACAAGTTTGTACAAAAAAGCAGGCTTAATGGACCAGTCTAAACAGGTTTTGC-3' | 5'-GGGGACCACTTTGTACAAGAAAGCTGGGTTCTATAGGCGCTCGGCCAAC-3' |
| Cnn-T-N fragment | 5'-GGGGACAAGTTTGTACAAAAAAGCAGGCTTAATGAATAGTAATCGAACGTCGTCTTCG-3' | 5'-GGGGACCACTTTGTACAAGAAAGCTGGGTTCTATAGGCGCTCGGCCAAC-3' |
| Cnn-C-N[P1] insert | 5'-GCGGGACTATTGCGGCGACGGCAATGGTACCTGTGCAGACGACTTGAAGGAAATCGAGTTAATTGAGGAGGTGG-3' | 5'-GCAGGACCCTTCTGTCGATTTCGGCGGCGCCATTCTCCTCCAGGAAGTCCTCCACCTCCTCAATTAACTCGATTTCC-3' |
| Cnn-C-N[P2] insert | 5'-CCTGCGCAAACTAGCCGAGGCACTGGACAAAGACATAGACGACGAGGACCCGGGAGCCCTGCAAGATGTCG-3' | 5'-CGCCTGGAGGTCGTGGAACGTCAAAGTCGGCATAGTCGTTCTCCATCTCGACATCTTGCAGGGCTCCCGGGTCC-3' |
| Cnn-C-N[P3] insert | 5'-GGGTCAGCCGGGTGCCCGGGCAGACGACGACGAGGAAGACTTAGACAAACAGCTCATCGATGCCAAGATCGAAATCGC-3' | 5'-CCTTGAGCAGCTCCATCTTTACATCGACCTCTTTTCTCAACTCCGCGATTTCGATCTTGGCATCGATGAGC-3' |
| Cnn-C-N[P1a] insert | 5'-GCGGGACTATTGCGGCGACGGCAATGGTACCTGTGCAGACGACTTGAAGGAAATCGAGTTAATTGAGACCGTGA-3' | 5'-GGACCCTTCTGTCGATTTCGGCGGCGCCATTCTCCTCCAGGAAACTGGTCACGGTCTCAATTAACTCGATTCCTTC-3' |
| Cnn-C-N[P1b] insert | 5'-GCGGGACTATTGCGGCGACGGCAATGGTACCTGTGCATCGTCCTTGAAGGAAATCACCTTAATTG-3' | 5'-GGACCCTTCTGTCGATTTCGGCGGCGCCATTCTCCTCCAGGAAGTCCTCCACCTCCTCAATTAAGGTGATTTCCTTC-3' |
| Cnn-C-N[T27] insert | 5'-GCGGGACTATTGCGGCGACGGCAATGGTACCTGTGCATCGTCCTTGAAGGAAATCGAGTTAATTGAGACCGTGA-3' | 5'-GGACCCTTCTGTCGATTTCGGCGGCGCCATTCTCCTCCAGGAAACTGGTCACGGTCTCAATTAACTCGATTTCCTTC-3' |
| Cnn-C-N[Δ1-77] | 5'-GCCAACTTTGTACAAAAAAGCAGGCTTAATGGCCAGTTTTGACGTTCC-3' | 5'-GGAACGTCAAAACTGGCCATTAAGCCTGCTTTTTTTGTACAAAGTTGGC-3' |
| CDK5RAP2 aa 1–210 | 5'-GGGGACAAGTTTGTACAAAAAAGCAGGCTTAATGATGGACTTGGTGTTGGAAGAGG | 5'-GGGGACCACTTTGTACAAGAAAGCTGGGTTTCACAAGTCCCCCTCGTGCATCTTC-3' |
| CDK5RAP2 aa 51–100 | 5'-GGGGACAAGTTTGTACAAAAAAGCAGGCTTAATGACAGTGTCTCCCACCAGAGCACG-3' | 5'-GGGGACCACTTTGTACAAGAAAGCTGGGTTTCAGTAGATATGTTCAGTGGG-3' |
| CDK5RAP2 aa 51–210 | 5'-GGGGACAAGTTTGTACAAAAAAGCAGGCTTAATGACAGTGTCTCCCACCAGAGCACG-3' | 5'-GGGGACCACTTTGTACAAGAAAGCTGGGTTTCACAAGTCCCCCTCGTGCATCTTC-3' |
| CDK5RAP2 aa 1–100 | 5'-GGGGACAAGTTTGTACAAAAAAGCAGGCTTAATGATGGACTTGGTGTTGGAAGAGG-3' | 5'-GGGGACCACTTTGTACAAGAAAGCTGGGTTTCAGTAGATATGTTCAGTGGG-3' |

4°C. Unbound fragments were washed off in PBS + 0.1% Tween 20 (PBST), and the saturated beads were resuspended in 100 μl embryo extract and rotated at 4°C overnight. Beads were washed five times for 1 min each in PBST, boiled in 50 μl 2× sample buffer, and separated from the sample using a magnet. Samples were analyzed by Western blotting as described.

For the Grip-GFP IPs, 20 μl high-capacity ProteinA beads (Abcam) coupled to anti-MBP antibodies (gift from Jordan Raff) were incubated with an excess of purified MBP-Cnn fragments and rotated at 4°C for 1 h. Unbound fragments were washed off in PBST and the saturated beads were resuspended in 65 μl embryo extract and rotated at 4°C overnight. Beads were washed five times for 1 min each in PBST, boiled in 2× sample buffer, and separated from the sample by centrifugation. Samples were analyzed by Western blotting as described.

For the immunoprecipitates from pUbq-Cnn-C and pUbq-Cnn-C[T] embryo extract, 50 μl magnetic ProteinA dynabeads (Life Technologies) coupled to anti-Cnn (C-terminal) antibodies (gift from Jordan Raff) were rotated in 100 μl embryo extract at 4°C overnight. Beads were washed five times for 1 min each in PBST, boiled in 2× sample buffer, and separated from the sample using a magnet. Samples were analyzed by Western blotting as described. We had tried these immunoprecipitates using beads coated with the anti-Cnn-T[N] antibody, but found that they did not pull down any protein

(data not shown), presumably as this antibody was raised against a peptide antigen and recognizes only denatured pUbq-Cnn-C[T] on Western blots.

**Electrophoresis and Western blotting**

Samples were run on 4–20% TGX Precast Gels (Bio-Rad; except Fig. 5, C and D, in which samples were run on 7.5% TGX Precast gels; Bio-Rad), alongside 5 μl Precision Plus WesternC Standard markers (Bio-Rad). For Western blotting, semi-dry blotting was performed using TransBlot Turbo 0.2-μm nitrocellulose membrane transfer packs (Bio-Rad) and a TransBlot Turbo transfer system running at 1.3 A, up to 25 V, for 7 min (Bio-Rad mixed molecular weight preset program). Membranes were stained with Ponceau and washed, first with distilled water then with milk solution (PBST + 4% milk powder), and then blocked in milk solution for 1 h at RT. Sections of blots were incubated with primary antibodies as indicated in figures (antibodies found in Table 2). Blots were incubated with HRP-conjugated anti-mouse, anti-rabbit, or anti-sheep secondary antibodies (1:2,000 in PBST + 4% milk powder; ImmunoReagents) as appropriate for 45 min at RT, washed in PBST three times for 15 min each, and then incubated with ECL substrate (Bio-Rad ECL Clarity or Thermo Fisher Scientific SuperSignal West Femto Max) for 5 min. Membranes were imaged using a Kodak Image Station 4000R or a Bio-Rad ChemiDoc.

Table 2. **Antibodies used in the study**

| Antibody | WB concentration | IF concentration | Source |
|---|---|---|---|
| α-Tubulin mouse monoclonal | — | 1:1,000 | Sigma-Aldrich; DM1a |
| Asl (N-terminal) guinea pig polyclonal | 1:1,000 | 1:1,000 | Gift from Jordan Raff |
| Cnn (N-terminal) rabbit monoclonal | 1:1,000 | 1:1,000 | Gift from Jordan Raff |
| Cnn (C-terminal) sheep polyclonal | 1:1,000 | — | Gift from Jordan Raff |
| Cnn-T$^N$ rabbit polyclonal | 1:500 | — | This study |
| γ-Tubulin mouse monoclonal | 1:500 | 1:500 | Sigma-Aldrich; GTU-88 |
| γ-Tubulin rabbit polyclonal | — | 1:500 | Sigma-Aldrich; T5192 |
| GFP mouse monoclonal | 1:250 or 1:500 | 1:250 or 1:500 | Roche, 11814460001 |
| Grip71 rabbit polyclonal | 1:100 | 1:100 | CRB (crb2005268) |
| MBP rabbit polyclonal | 1:3,000 | — | Gift from Jordan Raff |
| Phospho-histone H3 rabbit polyclonal | — | 1:500 | Abcam; AB5176 |
| Staufen mouse monoclonal | — | 1:100 | Santa Cruz Biotechnology; dN-16 |
| Gurken mouse monoclonal | — | 1:200 | DSHB; 1D12 |
| Lamin Dm0 | — | 1:30 | DSHB; 84.12 |

IF, immunofluorescence; WB, Western blotting.

## Mass spectrometry

Samples were run into TGX Precast Gels (Bio-Rad) and the gels were rinsed in dH$_2$O. Bands were excised using a clean razor blade and cut into 1-mm$^2$ pieces on a fresh glass slide and placed into a microtube. Co-IP samples were processed by the Mass Spectrometry facility at the Department of Biochemistry, University of Cambridge with liquid chromatography–tandem mass spectrometry analysis using a Dionex Ultimate 3000 RSLC nanoUPLC system (Thermo Fisher Scientific) and a Q Exactive Orbitrap mass spectrometer (Thermo Fisher Scientific).

After running, all tandem mass spectrometry data were converted to mgf files and the files were then submitted to the Mascot search algorithm (Matrix Science; version 2.6.0) and searched against the Uniprot Drosophila_melanogaster_20180813 database (23,297 sequences; 16,110,808 residues) and common contaminant sequences containing nonspecific proteins, such as keratins and trypsin (123 sequences; 40,594 residues). Variable modifications of oxidation (M), deamidation (NQ), and phosphorylation (S, T, and Y) were applied as well as a fixed modification of carbamidomethyl (C). The peptide and fragment mass tolerances were set to 20 ppm and 0.1 D, respectively. A significance threshold value of $P < 0.05$ and a peptide cutoff score of 20 were also applied.

## Antibodies

Primary antibodies used in the study are indicated in Table 2. For Western blotting, primary and secondary antibodies were diluted in PBST + 4% milk; primary antibodies were diluted at concentrations indicated in the table; secondary antibodies were diluted at 1:2000. For immunostaining, primary and secondary antibodies were diluted in PBS + 0.1% Triton X-100 (PBST) + 5% BSA. Primary antibodies were diluted at concentrations indicated in Table 2, and secondary antibodies (Alexa Fluor 488, -561, or -633–conjugated secondary antibodies; Thermo Fisher Scientific) were diluted at 1:1,000 for testes and 1:1,500 for embryos. DNA was stained with Hoechst (33342; Life Technologies) or DAPI.

## Immunostaining

Testes were dissected in PBS, fixed in 4% paraformaldehyde for 30 min, washed 3× for 5 min in PBS and incubated in 45% and then 60% acetic acid before being squashed onto slides and flash frozen in liquid nitrogen. Coverslips were removed and samples were postfixed in methanol at –20°C, washed 3× for 15 min in PBST, and then incubated overnight in a humid chamber at 4°C with primary antibodies diluted in PBST + 5% BSA + 0.02% azide. Slides were washed 3× for 5 min in PBST and then incubated for 2 h at RT with Alexa Fluor secondary antibodies (all 1:1,000 in PBST + 5% BSA + 0.02% azide; Thermo Fisher Scientific). Slides were washed 3× for 15 min in PBST, 10 min in PBST with Hoechst, and then 5 min in PBST. 10 µl of mounting medium (85% glycerol in water + 2.5% N-propyl-galate) was placed on top of the tissue and a coverslip was gently lowered and sealed with nail varnish.

Embryos were collected within 2–3 h of laying and were dechorionated in 60% bleach for 2 min. Vitelline membranes were punctured with a combination heptane and methanol + 3% EGTA (0.5 M) before three washes in neat methanol. Embryos were fixed in methanol at 4°C for at least 24 h before rehydrating. Embryos were rehydrated by washing 3× for 20 min in PBST and then blocked in PBST + 5% BSA for 1 h, followed by overnight incubation in primary antibodies in PBST + 5% BSA at 4°C. Embryos were washed 3× for 20 min in PBST at RT and then incubated for 2 h at RT with Alexa Fluor secondary antibodies (all 1:1,500 in PBST + 5% BSA; Thermo Fisher Scientific). Finally, embryos were washed 3× for 20 min in PBST at RT before being mounted in Vectashield containing DAPI (VectorLabs).

Oocytes were dissected from 2-d-old females. For Staufen and Gurken detection, 10–15 ovaries were fixed with PBS buffer containing 4% paraformaldehyde and 0.1% Triton X-100, washed 3× for 5 min in PBST and blocked in PBST containing 1% BSA. Incubation with the primary antibodies (anti-Staufen; Santa Cruz Biotechnology; anti-Gurken 1D12; Developmental Studies Hybridoma Bank) was performed overnight at RT or 4°C for Staufen and Gurken labeling, respectively, in PBST. Ovaries were then briefly washed three times and 3× for 30 min each in PBST and incubated for 2 h at RT in Alexa Fluor–conjugated secondary antibodies. The ovaries were then washed 3× for 15 min each in PBST, dissected, and mounted in Citifluor (Electron Microscopy Science).

### Preparation of testes for phase-contrast imaging of round spermatids

For analysis of round spermatids under phase contrast, testes were dissected in PBS, transferred to a 50-µl droplet of PBS on a slide, cut open midway along the testis, and, under observation, gently squashed under a coverslip using blotting paper.

### mRNA preparation and injection

pRNA vectors containing the appropriate cDNA were generated using Gateway cloning of PCR-amplified cDNA and either a pRNA-GFP or a pRNA-mKATE backbone. pRNA vectors were linearized with AscI; precipitated using EDTA, sodium acetate, and ethanol; and then resuspended in RNase-free water. mRNA was generated from these pRNA vectors in vitro using a T3 mMESSAGE mMACHINE kit (Thermo Fisher Scientific) and then purified using an RNeasy MinElute Cleanup kit (Qiagen). Freshly laid unfertilized eggs were collected from apple juice plates within ~1 h of laying and were dechorionated on double-sided sticky tape. Eggs were lined up on heptane glue to keep them in place during injections and imaging. Embryos were dried at 25°C for ~5 min and covered with immersion oil (Voltalef). mRNA was manually injected using a syringe into eggs using needles made from borosilicate glass capillary tubes at concentrations ranging from ~2–4 µg/µl. Eggs were left for 1.5–2 h before imaging to allow for translation of the mRNA. Control eggs were injected with RNase-free water.

### Fertility tests

We tested fertility rates of males and females bred at 25°C, comparing pUbq-Cnn-$C^T$ males or females to pUbq-Cnn-C males or females. We quantified the hatching rate of embryos that were generated when pUbq-Cnn-C or pUbq-Cnn-$C^T$ males or females were crossed to $w^{1118}$ WT flies. Cages that were sealed with apple juice agar plates with a spot of dried yeast paste were set up at 25°C containing ~50 newly hatched test flies (e.g., pUbq-Cnn-C/-$C^T$) and ~50 newly hatched WT males or virgin females. The apple juice agar plates were exchanged with fresh plates 2–4 times a day, and the removed plates were kept at 25°C for at least 25 h before the proportion of hatched eggs was calculated.

### Tissue culture, transfection, and immunoprecipitates from HEK cells

HEK293T cells were grown in high-glucose GlutaMAX DMEM supplemented with 10% heat-inactivated FBS and were incubated at 37°C and 5% $CO_2$. Cells were mycoplasma free (LookOut Mycoplasma PCR detection kit; Sigma-Aldrich). Cells were passaged with 0.05% trypsin-EDTA every 2–3 d. $7 \times 10^6$ cells were seeded and grown for 24 h before transfection. Cells were transfected with 1.45 µg DNA using Lipofectamine 2000 transfection reagent (Thermo Fisher Scientific) for 4 h in OptiMEM reduced-serum medium. A control flask was treated with Lipofectamine 2000 in the absence of any DNA but was otherwise processed identically. Medium was replaced with DMEM and cells were allowed to grow for a further 16 h before harvesting for immunoprecipitation.

Transfected cells were washed twice in PBS and lysed in buffer (50 mM Hepes [pH 7.5], 150 mM NaCl, 1 mM $MgCL_2$, 1 mM EGTA, 0.5% Octylphenoxy poly(ethyleneoxy)ethanol, branched [IGEPAL], and protease inhibitors), and then rotated for 90 min at 4°C. Cells were harvested at 15,000 rpm for 10 min at 4°C. The supernatant was mixed with 30 µl GFP-Trap_MA beads (Chromotek) and rotated overnight at 4°C. Beads were washed three times in ice-cold PBST and then resuspended in 50 µl 2× Laemmli sample buffer and boiled for 10 min at 95°C. Western blots were run as described above using anti-GFP (1:250; mouse; Roche) and anti–γ-tubulin (1:250; rabbit, T5192; Sigma-Aldrich) primary antibodies.

### Microscopy

All imaging was performed at RT (~20°C). Confocal imaging of fixed embryos—and the movies of scaffolds organizing microtubules—was performed on an Olympus FV3000 scanning inverted confocal system run by FV-OSR (Olympus Super Resolution) software using a 60× 1.4 NA silicone immersion lens (UPLSAPO60xSilicone) or 30× 0.95 NA silicone immersion lens (UPLSAPO30xSilicone). Confocal imaging of scaffolds recruiting γ-TuRC proteins or organizing microtubules as well as of testes samples was performed on a Zeiss Axio Observer.Z1 inverted CSU-X1 Yokogowa spinning disk system with 2 ORCA Fusion camera (Hamamatsu) run by Zeiss Zen2 acquisition software using a 60× 1.4 NA oil immersion lens (Zeiss). Confocal imaging of oocytes was performed on a Confocal LSM780 mount on a Axio Observer Z1 microscope (Zeiss), detection was done with spectral channels GaAsP, controlled by Zen software using a 25× 0.8 NA plan apochromat oil objective. Phase-contrast microscopy of round spermatids was performed on a Leica DM IL LED inverted microscope controlled by µManager software and coupled to a RetigaR1 monochrome camera (QImaging) using a 40× 0.55 NA air objective (Leica). Movies of scaffolds organising microtubules and spindle-like structures were recorded at a rate of 1 frame per 30 s.

### Image and statistical analysis

All images were processed using Fiji (ImageJ).

### *Quantifying and comparing the intensity of Cnn and γ-TuRC components at Cnn scaffolds*

Maximum-intensity Z-plane projections were made and a threshold mask was generated using the Cnn channel. Sum fluorescence intensities for the Cnn and γ-TuRC protein channels were calculated. Overall mean cytosolic background

intensity measurements for each channel were used to background correct the sum intensities for each scaffold. The scaffold intensities within each egg were plotted in Prism and a weighted linear regression analysis—based on ensuring an even distribution of residuals across the x-axis—was performed. The gradient of the weighted regression line represented the S value for a given egg. The distribution of the S values per condition were lognormally distributed—as determined by Anderson-Darling, D'Agostino and Pearson, Shapiro-Wilk, and Kolmogorov-Smirnov tests—and so, to compare mean S values, the $\log_{10}$ of each individual S value was first calculated before performing a one-way ANOVA analysis (Dunnett's multiple comparisons test). This was to ensure that the data being compared were normally distributed. Nevertheless, the unadjusted S values were plotted. Note that the fluorescence values and S values are in arbitrary units and cannot be directly compared between the γ-tubulin-mCherry and Grip75-sfGFP analysis.

### Blind analysis of Cnn scaffolds organizing microtubules

Images were blinded and saved with the same contrast settings. Images were then selected on scaffold size, with eggs containing small or very large scaffolds removed. The remaining images were then scored by eye as being of eggs that contained scaffolds with either no asters, weak or strong asters, or scaffolds where the Jupiter-mCherry signal did not extend beyond the GFP-Cnn signal (overlay).

### Blind analysis of pUbq-Cnn-C or pUbq-Cnn-C$^T$ embryos

Images were blinded and saved with the same contrast settings. Embryos were then scored by eye as being either normal or having moderate or severe defects. Embryos were scored as normal even when one or two mitotic figures had defects, because this is quite common in syncytial embryos and does not prevent development. Embryos were scored as having moderate defects when an unusually high proportion of mitotic figures had defects or where the overall organization of the spindles was moderately abnormal. Embryos were scored as having severe defects when there was either massive disorder with individual mitotic figures or overall organization or both.

### Quantifying Western blot bands

The sum intensities of bands were background corrected using mean background values at positions on the gel with no apparent signal. To reduce variation, the band intensities were taken using the freehand tool to draw closely around the perimeter of the band. For co-IP experiments, the intensities of the γ-tubulin immunoprecipitate bands were normalized to the intensity of the γ-tubulin band in the MBP-Cnn-T-N IP within each experiment. GraphPad Prism 7 or 8 was used for all statistical analysis and graph production.

### Bioinformatics

Protein alignments were produced using JalView. Secondary structure predictions were performed using JPred 4.

### Online supplemental material

Fig. S1 shows the different constructs that were used in the study. Fig. S2 shows that MBP-Cnn-T-N fragments coimmunoprecipitate various γ-TuRC proteins. Fig. S3 shows alignments of the N-terminal region of Cnn-C from different *Drosophila* species and with homologues from more diverse species. Fig. S4 shows images of Staufen and Gurken localization within pUbq-Cnn-C or pUbq-Cnn-T oocytes. Fig. S5 shows images of spermatocytes stained for DNA, centrosomes, and microtubules within pUbq-Cnn-C or pUbq-Cnn-T testes. Video 1 shows Cnn-T scaffolds organizing microtubule asters. Video 2 shows transient spindle-like structures forming between (and disappearing from) Cnn-T scaffolds. Video 3 shows spindle-like structures organized by Cnn-T scaffolds forming in synchrony. Video 4 dislays rare giant Cnn-T scaffolds to which microtubules are robustly anchored. Video 5 shows dynamic cytosolic microtubules within eggs injected with mRNA encoding GFP-Cnn-T-N.

## Acknowledgments

We thank Jordan Raff for sharing reagents, Berthold Hedwig and Steve Rogers for help with needle pulling, Jens Lüders for sharing the pCMV-GFP vector, and Matt Castle for guidance on statistical analysis. We thank other members of the Conduit laboratory for their invaluable input. The work benefited from the Imaging Facility, Department of Zoology, University of Cambridge, supported by Matt Wayland and a Sir Isaac Newton Trust Research Grant (18.07ii[c]) from the ImagoSeine at the Institut Jacques-Monod, and from use of the Cambridge Centre for Proteomics Core Facility. For the purpose of open access, the author has applied a CC-BY public copyright license to any Author Accepted Manuscript (AAM) version arising from this submission.

This research was supported by a Wellcome Trust and Royal Society Sir Henry Dale fellowship (105653/Z/14/Z) and by IdEx Université de Paris (grant ANR-18-IDEX-0001 to P.T. Conduit), by a Glover Fund research fellowship from Clare College, University of Cambridge (to C.A. Tovey), by an Association pour la Recherche sur le Cancer grant (PJA 20181208148 to A. Guichet), and by the Centre National de la Recherche Scientifique.

The authors declare no competing financial interests.

Author contributions: C.A. Tovey: funding acquisition, conceptualization, formal analysis, investigation, supervision, validation, writing - review and editing; C. Tsuji: formal analysis and investigation; A. Egerton: investigation; M. de la Roche: investigation and resources; A. Guichet: resources, investigation, formal analysis, and validation; F. Bernard: resources; and P.T. Conduit: funding acquisition, conceptualization, project administration, formal analysis, investigation, resources, supervision, validation, visualization, writing - original draft and review and editing.

Submitted: 5 October 2020

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

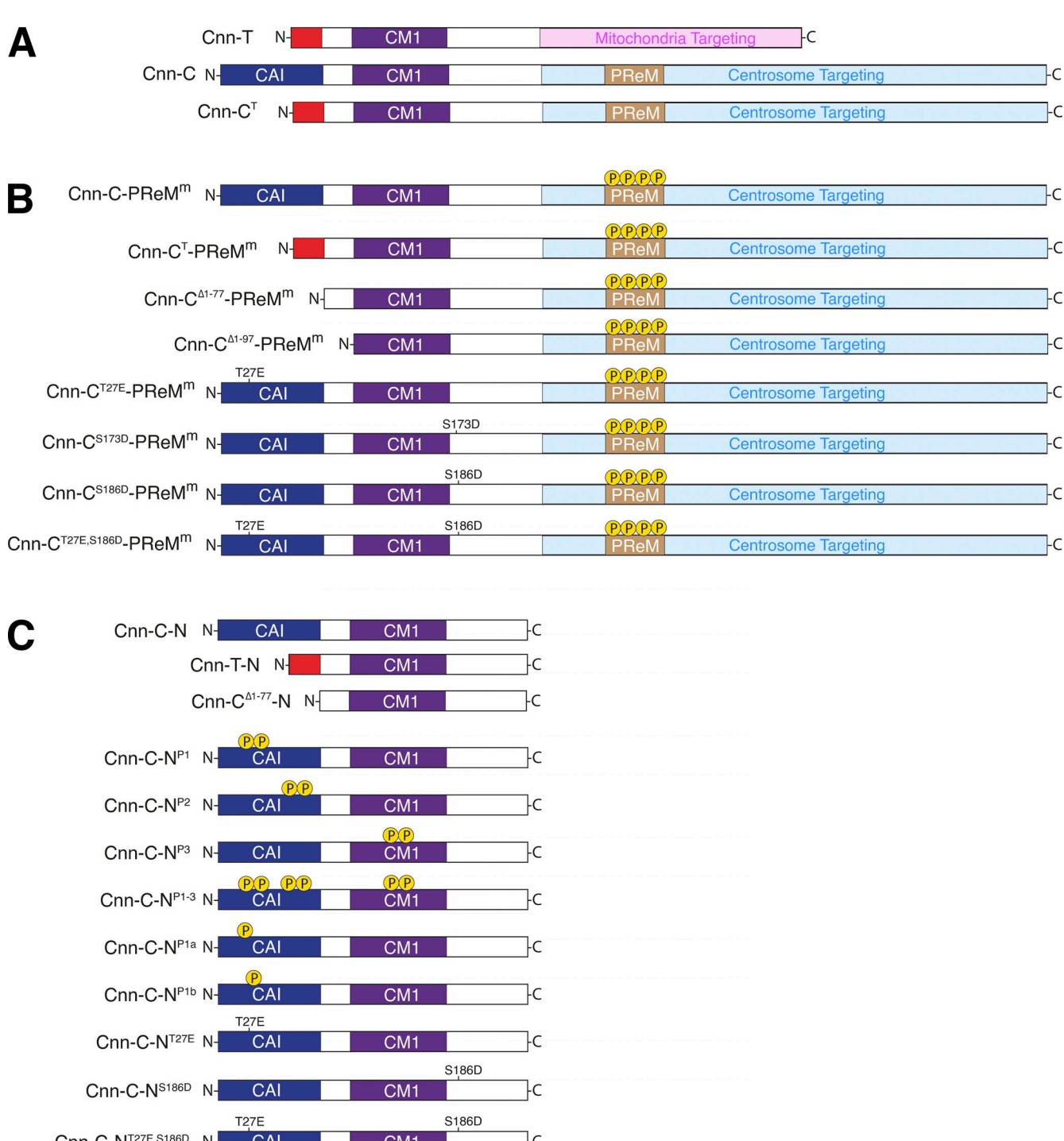

Figure S1. **Diagrams of different Cnn constructs (omitting the tags) used in this study. (A)** Diagram showing full-length Cnn constructs without modifications to the PReM domain. Cnn-T is the testes-specific isoform in *Drosophila*. Cnn-C is the major centrosomal isoform in *Drosophila*. Cnn-C$^T$ represents an artificial form of Cnn-C in which the N-terminal region of Cnn-C (dark blue) has been replaced with the N-terminal region of Cnn-T (red). **(B)** Diagram showing Cnn constructs used in the scaffold assay, where Cnn-C contains phosphmimetic mutations in the PReM domain to drive scaffold formation in vivo. **(C)** Diagram showing bacterially purified N-terminal fragments of different Cnn types used in co-IP experiments.

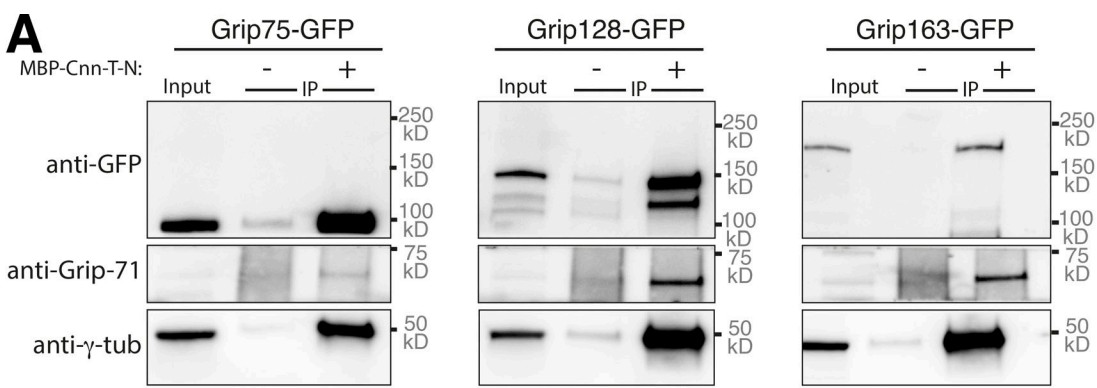

**B**

| UniProt entry | Protein | G75, no fragment | G75 + Cnn-T-N | G128 + Cnn-T-N | G163 + Cnn-T-N |
|---|---|---|---|---|---|
| A0A0B4K6Z9 | **Cnn** | - | 4.50 | 15.63 | 13.03 |
| M9PDN9 | **γ-tubulin 37C** | 0.45 | 24.45 | 63.16 | 39.41 |
| P23257 | **γ-tubulin 23C** | - | 1.04 | 1.90 | 1.43 |
| E1JJQ3 | **Grip84** | - | 1.76 | 8.14 | 4.76 |
| Q8IQW7 | **Grip84** | - | 1.54 | 7.44 | - |
| Q9XYP8 | **Grip91** | - | 1.74 | 4.45 | 3.54 |
| Q9VKU7 | **Grip75** | 0.07 | 0.88 | 3.57 | 2.54 |
| Q9VXU8 | **Grip128** | - | 0.08 | 0.40 | 0.16 |
| Q9VTS3 | **Grip163** | - | 0.24 | 1.09 | 0.24 |
| Q9VJ57 | **Grip71** | - | 0.30 | 1.87 | 0.93 |
| X2JCP8 | **Actin** | 40.63 | 17.87 | 22.66 | 32.21 |

Figure S2. **Bacterially purified MBP-Cnn-T-N fragments coimmunoprecipitate γ-tubulin ring complexes. (A)** Western blot showing results of anti-MBP co-IP from embryo extracts expressing GFP-tagged Grip proteins (homologues of GCP4,5,6), either supplemented (+) or not supplemented (–) with MBP-Cnn-T-N, as indicated. Blots were probed with anti-GFP, anti-Grip71, and anti–γ-tubulin antibodies as indicated. When using MBP-Cnn-T-N, γ-tubulin, and Grip71, as well as Grip75, 128, or 163, are coimmunoprecipitated. **(B)** Mass spectrometry results from IPs with MBP-Cnn-T-N showing the presence of various γ-TuRC components. Note that Mzt1 is not expressed within embryos. Results of a control experiment on Grip75-GFP embryo extract not supplemented with any MBP-Cnn-T-N fragment are also shown. Numbers indicate emPAI scores as a proxy for protein abundance. Grip84 (A) and Grip84 (E) represent two different isoforms of Grip84 (promoters 1 and 2, respectively).

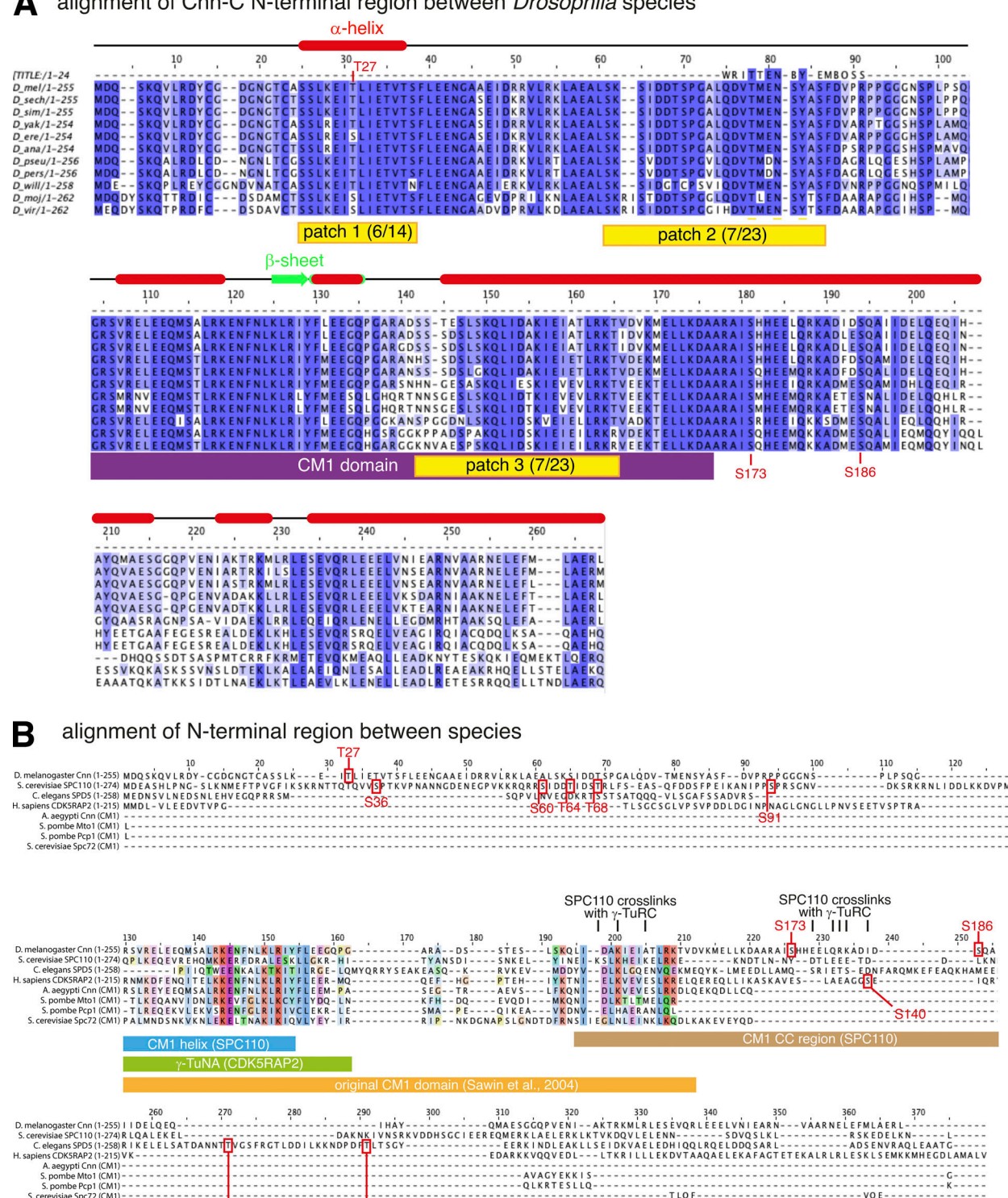

Figure S3.   **Protein alignments of N-terminal regions of CM1 domain proteins. (A)** An alignment of Cnn-C homologues from different *Drosophila* species. The alignment was performed in JalView keeping *Drosophila melanogaster* at the top with the closest related species in order below. Only the N-terminal regions of the proteins were used in the alignment (~1–255 aa). Potential phosphorylation patches are highlighted in yellow, with the proportion of S/T residues present in the *D. melanogaster* sequence indicated in brackets. The CM1 domain is highlighted in purple. Red boxes and green arrows indicate α-helices and β-sheets based on predictions from JPred. **(B)** An alignment of the N-terminal region of Cnn-C with the equivalent N-terminal regions of its homologues in non-*Drosophila* species. Phosphorylation sites that promote binding to γ-TuRCs identified either in this study (*Drosophila* Cnn-C, T27 and S186) or other studies (*S. cerevisiae* SPC110 S36, S60, T64, T68 and S91, *C. elegans* SPD-5, T178, T198; human CDK5RAP2, S140) are indicated. Note that only the originally identified CM1 domain sequence (yellow) is conserved between homologues. The position of the CM1 helix (blue) and CM1 coiled-coil (CC) region (brown) recently identified in SPC110 are indicated, as is the γ-TuNA sequence from human CDK5RAP2 (green).

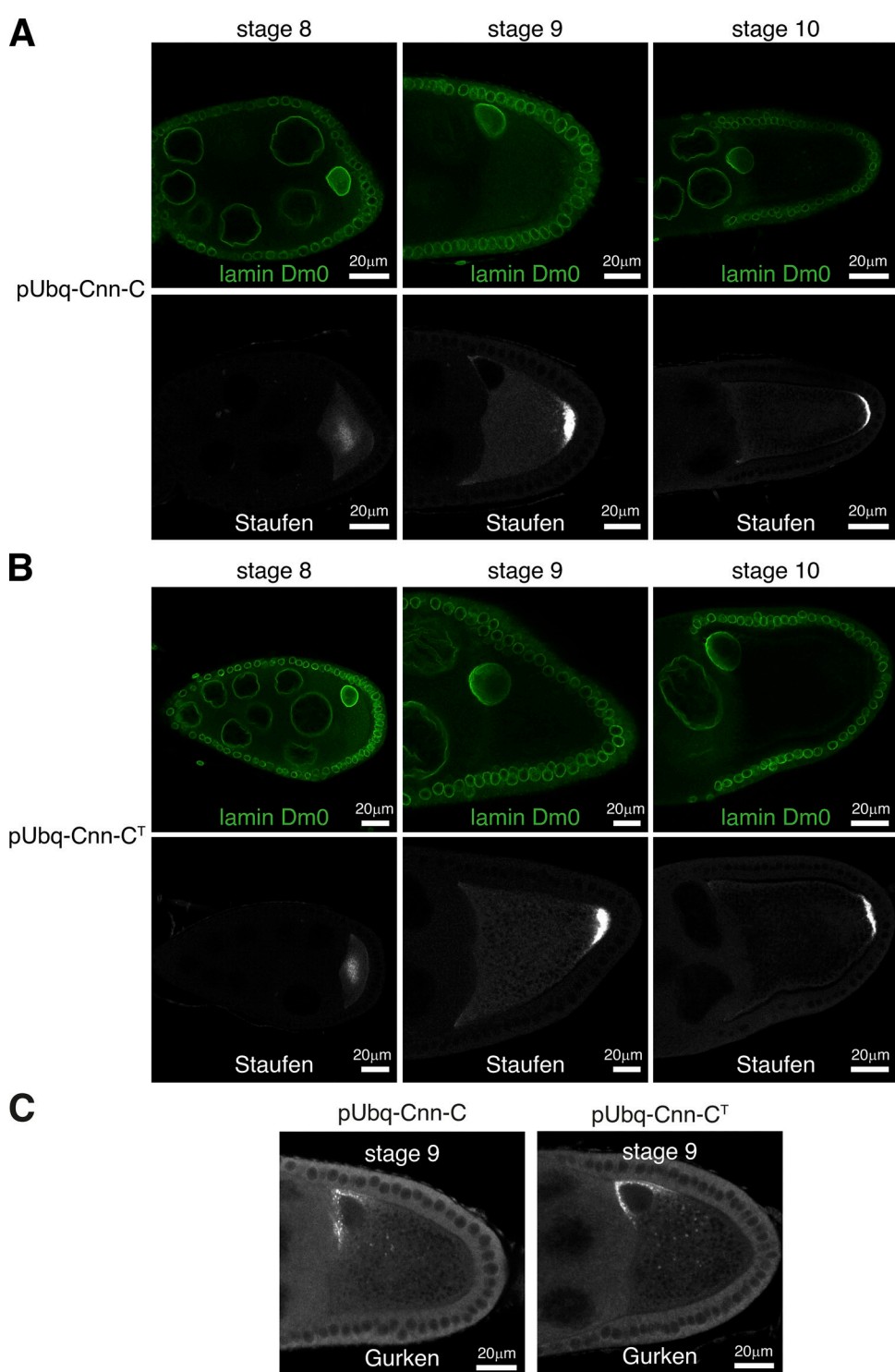

Figure S4. **Polarity is established normally in pUbq-Cnn-C$^T$ oocytes. (A and B)** Fluorescence images show localization of Staufen protein in oocytes expressing pUbq-Cnn-C (A) or pUbq-Cnn-C$^T$ (B) at stages 8, 9, and 10, as indicated. Staufen localized in the center of the oocyte at stage 8 and then at the posterior in stage 9 and 10 in all pUbq-Cnn-C ($n$ = 35, stage 8; $n$ = 35, stage 9; $n$ = 30, stage 10) and all pUbq-Cnn-C$^T$ ($n$ = 40, stage 8; $n$ = 50, stage 9; $n$ = 40, stage 10) oocytes that were imaged. **(C)** Fluorescence images show localization of Gurken protein in oocytes expressing pUbq-Cnn-C or pUbq-Cnn-C$^T$ at stage 9. Gurken protein was positioned close to the nucleus in the dorsal corner in all pUbq-Cnn-C ($n$ = 30) and all pUbq-Cnn-C$^T$ ($n$ = 35) stage 9 oocytes. Gurken mispositioning or its absence results in abnormal dorsal appendages that protrude from the surface of the egg, but the dorsal appendages were normal on all pUbq-Cnn-C ($n$ = 724) and all pUbq-Cnn-C$^T$ ($n$ = 488) eggs.

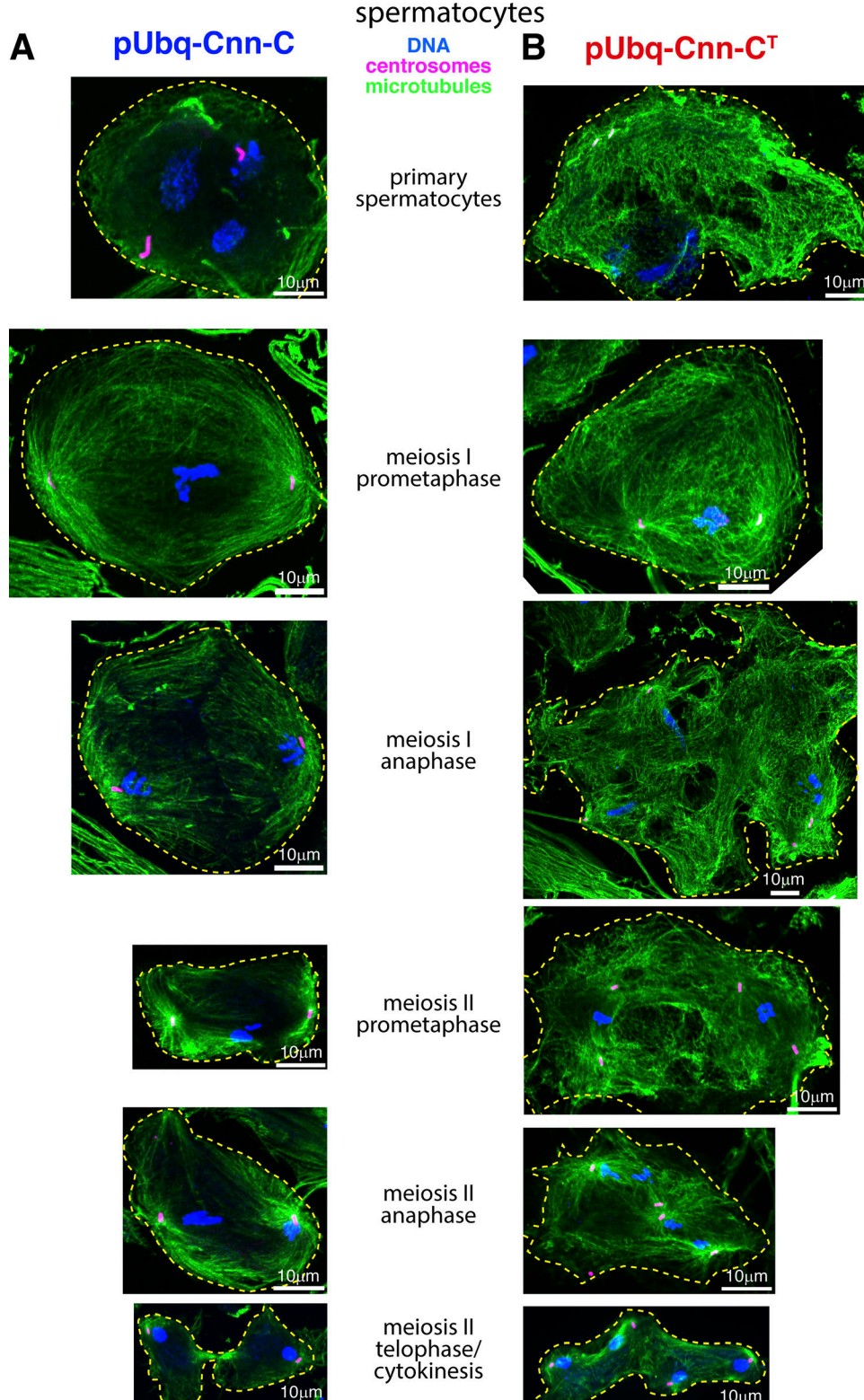

**Figure S5. Major spermatocyte defects are observed within testes from pUbq-Cnn-C$^T$ flies. (A and B)** Fluorescence images showing spermatocytes at different developmental stages (as indicated) from flies expressing either pUbq-Cnn-C (A) or pUbq-Cnn-C$^T$ (B) fixed and stained for microtubules (green, α-tubulin), centrosomes (pink, asterless), and DNA (blue). A high density of cytosolic microtubules, as well as cytokinesis and karyokinesis defects, are clearly observed in cells from pUbq-Cnn-C$^T$ testes.

Video 1.   **Cnn-T scaffolds organize microtubule asters and can be mobile.** Movie showing Cnn-T scaffolds (green) organizing microtubule asters (marked with Jupiter-mCherry; magenta). A mobile scaffold (lower left) with an asymmetric microtubule aster can be seen moving through the cytosol.

Video 2.   **Transient spindle-like structures can form between Cnn scaffolds.** Movie showing the formation and disappearance of a transient spindle-like structure between adjacent Cnn-T scaffolds (green). Microtubules are marked with Jupiter-mCherry (magenta).

Video 3.   **Spindle-like structures organized by Cnn scaffolds can form in synchrony.** Movie showing the synchronous formation and disappearance of a multipolar spindle-like array of microtubules that is subsequently organized by a nearby group of coalescing Cnn scaffolds (green). Microtubules are marked with Jupiter-mCherry (magenta).

Video 4.   **Microtubules are robustly anchored to Cnn scaffolds.** Movie showing rare giant Cnn-T scaffolds (green). One scaffold can be seen rotating and dragging the microtubules, indicating that the microtubules are robustly attached to the scaffold, presumably via γ-TuRCs. Microtubules are marked with Jupiter-mCherry (magenta).

Video 5.   **Expression of GFP-Cnn-T-N leads to the formation of dynamic microtubules within the cytosol of unfertilized eggs.** Video shows the effect of injecting mRNA encoding GFP-Cnn-T-N into unfertilized eggs expressing Jupiter-mCherry (marker of microtubules). Left panel shows the GFP channel (green), center panel shows the RFP channel (magenta), and right panel shows a merge.

