## [Peer Review File · The Journal of Cell Biology]

Auto-inhibition of Cnn binding to γ -TuRCs prevents ectopic MT nucleation and cell division defects

Corinne Tovey, Chisato Tsuji, Alice Egerton, Fred Bernard, Antoine Guichet, Marc de la Roche, and Paul Conduit

Corresponding Author(s): Paul Conduit, Institut Jacques Monod

Review Timeline:

Submission Date:	2020-10-05
Editorial Decision:	2020-11-09
Revision Received:	2021-03-25
Editorial Decision:	2021-04-21
Revision Received:	2021-04-27

Monitoring Editor: Tarun Kapoor

Scientific Editor: Andrea Marat

Transaction Report:

DOI: <https://doi.org/10.1083/jcb.202010020>

November 9, 2020

Re: JCB manuscript #202010020

Dr. Paul T Conduit
Institut Jacques Monod
15 rue Hélène Brion
Paris 75205
France

Dear Dr. Conduit,

Thank you for submitting your manuscript entitled "Phospho-regulated auto-inhibition of Cnn controls microtubule nucleation during cell division". The manuscript was assessed by expert reviewers, whose comments are appended to this letter. We invite you to submit a revision if you can address the reviewers' key concerns, as outlined here.

As you will see, the reviewers find the identification of an autoinhibitory regulatory mechanism for Cnn a potentially important finding for the field. However, they have all commented that further experimental evidence is necessary to validate the binding and the underlying molecular details regarding this regulatory mechanism, therefore in revising please thoroughly address all of their constructive comments pertaining to these points with new experimental evidence where requested. However, the comments about condensates (reviewer 2 point 1), as well as the concerns regarding results in embryos all seem less essential to address experimentally. Please also ensure that you sufficiently tone down any conclusions not supported by the data in your revised manuscript accordingly.

GENERAL GUIDELINES:

Text limits: Character count for an Article is < 40,000, not including spaces. Count includes title page, abstract, introduction, results, discussion, acknowledgments, and figure legends. Count does not include materials and methods, references, tables, or supplemental legends.

Figures: Articles may have up to 10 main text figures. Figures must be prepared according to the policies outlined in our Instructions to Authors, under Data Presentation, <https://jcb.rupress.org/site/misc/ifora.xhtml>. All figures in accepted manuscripts will be screened prior to publication.

Supplemental information: There are strict limits on the allowable amount of supplemental data. Articles may have up to 5 supplemental figures. Up to 10 supplemental videos or flash animations are allowed. A summary of all supplemental material should appear at the end of the Materials and methods section.

As you may know, the typical timeframe for revisions is three to four months. However, we at JCB realize that the implementation of social distancing and shelter in place measures that limit spread of COVID-19 also pose challenges to scientific researchers. Lab closures especially are preventing scientists from conducting experiments to further their research. Therefore, JCB has waived the revision time limit. We recommend that you reach out to the editors once your lab has reopened to decide on an appropriate time frame for resubmission. Please note that papers are generally considered through only one revision cycle, so any revised manuscript will likely be either accepted or rejected.

Thank you for this interesting contribution to Journal of Cell Biology. You can contact us at the journal office with any questions, cellbio@rockefeller.edu or call (212) 327-8588.

Sincerely,

Tarun Kapoor, PhD
Monitoring Editor

Andrea L. Marat, PhD
Senior Scientific Editor

Journal of Cell Biology

Reviewer #1 (Comments to the Authors (Required)):

In this manuscript, the authors analyzed biological functions of the N-terminal domain of *Drosophila* Cnn. They identified an extreme N-terminal "CM1 auto-inhibition (CAI)" domain within the centrosomal isoform of Cnn (Cnn-C) that inhibits γ -TuRC binding. The testes-specific mitochondrial Cnn-T isoform lacks the CAI domain and can bind strongly to cytosolic γ -TuRCs. The phosphomimetic mutations within CAI enhanced γ -TuRC binding, suggesting that the CAI activity is regulated by phosphorylation. Based on the results, they propose a model that the CAI domain folds back to sterically inhibit the CM1 domain, and that this auto-inhibition is relieved by phosphorylation that occurs specifically at centrosomes.

A key discovery of the manuscript may be that the Cnn interaction with γ -TuRCs is regulated by CAI which is located at the N-terminal end. This finding is significant. However, the current experiments do not support their conclusion sufficiently. Some of their works are indirect. Additional works are required be published in JCB. Follows are specific points.

- Figures 2, 3 5: There is no reason not to include Cnn-CD1-77 and Cnn-C-N P1 in the analyses.
- Figure 3: The amounts of Cnn-C at the spindle poles look smaller than those of Cnn-T. It may be worth to compare the aster intensities of Cnn-C and Cnn-T with their spindle pole intensities.
- Figure 5E: The authors may explain why a half of the wild type embryos had defects.
- Figure S3: The putative CAI sequence may be compared with the *C. elegans* and human homologues. Furthermore, experiments in Figure 4A-D may be performed with human CDK5RAP2.
- Since CAI inhibits γ -TuRC binding to CM1, it may physically interact with CM1. The physical interaction could be regulated by phosphorylations at specific residues. This prediction may be examined with the GST pulldown and/or coimmunoprecipitation assays with the CAI and CM1 fragments.

Reviewer #2 (Comments to the Authors (Required)):

The ability of microtubules to be built with spatiotemporal precision in cells relies on the activity of microtubule organizing centers (MTOCs). During mitosis, the centrosome acts as an MTOC to recruit PCM proteins that in turn build microtubules to establish the mitotic spindle. Despite the ubiquity of this process across metazoan cell types, the intramolecular interactions that drive microtubule assembly at the centrosome during mitosis are not well understood. Here, Tovey et al. establish an *in vivo* g-tubulin complex recruitment assay in *Drosophila* to identify a critical region of Centrosomin/CDK5RAP2 that is central to g-TuRC and microtubule recruitment. Through a combination of *in vivo* genetic and imaging and *in vitro* biochemical experiments, the authors propose a model where the N-terminal region of Cnn is autoinhibited in a manner that blocks g-tubulin complex recruitment. Phosphorylation by Polo relieves this inhibition, allowing g-tubulin complex recruitment and subsequent microtubule nucleation. While the data presented are rigorous and clear and the model is simple and elegant, there are a few glaring holes in the data that would need to be addressed to actually support this model.

Major Comments:

- 1) Central to the authors' model, the CAI inhibits the ability of Cnn to interact with g-tubulin complexes. This conclusion is clear from both the *in vivo* and *in vitro* data. Similarly, it is clear from the data that the "uninhibited" form of Cnn can drive assembly of microtubules *in vivo*. Given that condensates of the Cnn ortholog SPD-5 have been shown to be able to induce microtubule growth independent of g-TuRC, I think it is crucial to show that the ability of Cnn condensates to build microtubules in this *in vivo* assay is g-tubulin dependent. This could be done through g-tubulin depletion experiments combined with the expression of CnnC or CnnT in eggs.
- 2) Another key point in the authors' model is that phosphorylation of the CAI by Polo relieves the inhibition of this region and allows for g-tubulin complex binding. This connection is weak at best. The authors demonstrate that phosphomimetics have increased g-tubulin binding *in vitro*, however, they show neither that these phosphorylation sites matter *in vivo* nor that they are regulated by Polo. Potential experiments to test either of these claims are referenced in the Discussion, but the data is not shown and despite having negative data the authors do not change the claims of their

model. The language needs to be significantly softened throughout and experiments should be presented as Supplemental data to make it clear that the current data do not support this part of the model.

3) The authors claim that the ability to regulate microtubule growth in time and space matters to the cell and organism. While the sperm defects from CnnT overexpression are clear, the defects presented in the embryo are not. In particular, in Figure 5E I find it alarming that the wild-type control in this experiment has such a significant proportion of mitotic defects. The fact that the control is so perturb makes me question whether the phenotypes seen in CnnC and CnnT eggs are due to the specific effect of these proteins on the process in question vs. a synthetic interaction with background markers. In the discussion, the authors note that spurious cytoplasmic microtubule nucleation "major defects during cell division" (line 503), which is an overstatement based on the control in this case. Is this indeed a background effect? Are the same effects seen in CnnC or CnnT eggs that have been fixed and stained instead of expressing other endogenously tagged proteins?

Minor comments:

- Line 126: missing character
- Figure 5F is not referenced in the main Results section
- Define Nebenkern
- Line 489: nucleat[ed]

Reviewer #3 (Comments to the Authors (Required)):

The authors identify an inhibitory domain that is present in the amino-terminal region of centrosomin (Cnn-C), but absent from a testis-specific Cnn isoform (Cnn-T). By performing domain swapping with the amino-terminal region of Cnn-T, or by removing the first 77 amino acids from Cnn-C, they create scaffolds of Cnn that bind constitutively high amounts of gamma-TuRCs when overexpressed in *Drosophila* eggs. The scaffolds formed from Cnn-T are able to organize microtubule asters much more efficiently than Cnn-C scaffolds. Expressing Cnn-T ubiquitously in flies leads to mitotic defects, defects in development, and less progeny. The authors propose that an inhibitory amino-terminal region in Cnn-C folds back onto the region containing the CM1 motif and thereby prevents binding of gamma-TuRCs to Cnn. Phosphomimetic mutations of different patches in the amino-terminal region (termed P1 and P2) relieve autoinhibition of Cnn, and restore partially the binding of gamma-tubulin to Cnn in *Drosophila* embryonic extracts.

Altogether, the identification of a regulatory mechanism of Cnn by autoinhibition would be an immensely important information for a wide readership. However, I think that the manuscript in its present form is not focused enough, and the idea of autoinhibition by folding back of the Cnn amino-terminus would benefit from additional experiments:

1) The main message of the manuscript is on the autoinhibitory function of the Cnn amino-terminus. The data on hatching and embryonic defects in *Drosophila* are of lower priority, as long as the cellular mechanisms aren't documented more clearly. I think the authors are skipping steps here. Whereas it is well documented that artificial Cnn-T scaffolds attract gamma-TuRCs and induce microtubule asters (Figures 2 and 3), it would be helpful to have clearer pictures of mitotic spindles and Cnn-CT-induced spindle defects in embryos, and well-resolved images on the formation of supernumerary microtubule-organizing centres (Figure 5). The immunofluorescence images in Figures 5F and 5I are too small and too fuzzy to make a strong point; and Figure S5 should show control cells (Cnn-C or wild type) for comparison.

2) The autoinhibitory action of the Cnn-C amino-terminal region should be tested more rigorously, with additional experiments: can the authors produce protein fragments in bacteria containing P1, P2, P1+2, CM1+P3, and test directly whether the P1-containing fragments bind to the CM1 region? This would be important as direct proof of their model in Figure 4A, suggesting the folding-back of P1+P2 onto Cnn.

Other points:

- Lines 199-216 and Figure 2: The S values for Grip75, Grip 128, and Grip163 vary from gamma-Tubulin37C, and the authors speculate that this might be due to the lower abundance of these three GCPs within the gamma-TuRC. But why are the values for Grip128 and Grip163 so different from each other, although both proteins are present in the gamma-TuRC at the same stoichiometry?

- I think that Figure 3D and the corresponding comments in the 'results' section are unnecessary and are distracting from the main message of the manuscript (as long as the spindle defects in cells/embryos aren't documented more clearly).

Dear Reviewers,

We thank the Reviewers and Editors for their thoughtful and constructive comments. We have carefully considered them all and directly addressed as many as possible. It was clear from the comments that the Reviewers felt the main message of the paper (that Cnn's binding to γ -TuRCs is auto-inhibited by the N-terminal CAI domain) is important, but that we needed to fill in some gaps and either add new experimental data or tone down some of our conclusions regarding the precise molecular details governing the autoinhibition mechanism.

Covid-19 has of course increased the time taken to submit this revised version and we appreciate the Reviewers' patience. Another factor that has delayed submission is that we have had to re-acquire and re-analyse most of the original imaging data (scaffold images within eggs and testes images). This is because we no longer have access to the microscopes used in the original paper (the lab recently moved from Cambridge to Paris) and we needed to make comparisons between the original data and the new data requested by the Reviewers.

To summarise briefly the main additions to the paper, we have obtained more data to support the conclusion that phosphorylation helps relieve auto-inhibition and promote binding to γ -TuRCs. We have now used the *in vivo* scaffold assay to test various phospho-mimetic mutations (Figure 4G-I) and this has helped to confirm a role for T²⁷ and also led to the discovery of another putative phosphorylation site, S¹⁸⁶. Mimicking S¹⁸⁶ appears to relieve auto-inhibition fully, at least in the co-IP assay, and phospho-mimicking T²⁷ and S¹⁸⁶ is synergistic, leading to extremely strong binding (Figure 4E). The position of S¹⁸⁶, downstream of the CM1 domain, is similar to the position of sites within human CDK5RAP2 (Hanafusa et al., 2015) and *C. elegans* SPD-5 (Ohta et al., 2021) that promote binding to γ -TuRCs, suggesting that phosphorylation in this region is a conserved feature of CM1 domain proteins. In addition, we find that human CDK5RAP2 is also auto-inhibited from binding γ -TuRCs by the region downstream of the CM1 domain (Figure 8). While the mechanism of auto-inhibition may be slightly different, our data suggest that auto-inhibition is a conserved feature of CM1 domain proteins.

While we have made progress, we still do not have a complete picture of how phosphorylation regulates auto-inhibition and so we have changed the title of the manuscript to: “**Auto-inhibition of Cnn binding to γ -TuRCs prevents ectopic microtubule nucleation and cell division defects**”. We feel this title better reflects the main message of the paper, particularly because we have now more clearly presented the data on cell and developmental defects.

There are now 8 main figures, rather than 5, in order to accommodate the new data and analysis. Note also that the text has been made more concise in order to reduce the character count below 40,000, as requested by the Editors.

Below we explain how we have addressed each point in turn. The original comments are in grey, our responses are in black and any new text from the revised manuscript that we quote here is in orange. We have numbered the comments for clarity.

Reviewer #1 (Comments to the Authors (Required)):

1. Figures 2, 3 5: There is no reason not to include Cnn- Δ 1-77 and Cnn-C-N P1 in the analyses.

Reviewer 1 is referring to the original figures that compared Cnn-C (low binding) to Cnn-T/Cnn-C^T (high binding) when examining recruitment of GFP-tagged γ -TuRC-specific proteins to Cnn scaffolds (Figure 2), the ability of Cnn scaffolds to organise microtubules (Figure 3), and the effect of expressing different Cnn constructs *in vivo* (original Figure 5). We believe the Reviewer might

have also wanted to include Figure 1 here (analysing the recruitment of γ -tubulin-mCherry to scaffolds), where we had analysed Cnn-C Δ^{1-77} scaffolds but not scaffolds with phospho-mimetic mutations in the N-terminal region.

We have therefore now analysed Cnn-C Δ^{1-77} scaffolds and scaffolds with N-terminal phospho-mimetic mutations in the various scaffold assays (γ -tubulin recruitment, Grip75 recruitment, and microtubule organisation). The data for the phospho-mimetic scaffold analysis is presented in Figure 4 to maintain the logical flow of the paper. Note that we had originally analysed the recruitment of 3 γ -TuRC-specific proteins (Grip75, Grip128, Grip163) to Cnn-C vs Cnn-T scaffolds in Figure 2. Given that we have now analysed more scaffold types we decided to focus the analysis on Grip75, especially given that our original data suggested that it was a better marker, presumably because of its higher stoichiometry in the complex (2 vs 1 molecule per γ -TuRC). Note, however, that we still include the IP and mass spectrometry data showing that Cnn-T-N fragments bind to complexes containing all other γ -TuRC proteins (Figure S2).

To summarise the results, Cnn-C Δ^{1-77} scaffolds recruit γ -tubulin, Grip75, and organise microtubules similar to Cnn-T scaffolds (Figures 1-3). Rather than analysing Cnn-C-N P1 scaffolds, we focused on scaffolds with individual or dual phospho-mimetic mutations. We found that scaffolds with a phospho-mimic mutation at T²⁷ (the putative Polo/Plk1 site within P1 that had increased binding *in vitro*) recruited γ -tubulin better than Cnn-C scaffolds, although not as well as either Cnn-T or Cnn-C Δ^{1-77} scaffolds. We therefore searched for other sites, encouraged partly by previous findings in humans and *C. elegans* that sites downstream of the CM1 domain promote binding to γ -TuRCs. We found that scaffolds with phospho-mimic mutations at S¹⁸⁶ recruited γ -tubulin better than Cnn-C and Cnn-C^{T27} scaffolds, although again not as well as either Cnn-T or Cnn-C Δ^{1-77} scaffolds. We believe this may be because the phospho-mimetic scaffolds seem to only recruit γ -TuSCs, as Grip75-GFP was not recruited. Strikingly, however, the phospho-mimetic mutations led to very strong γ -TuRC binding *in vitro*. N-terminal fragments containing the S¹⁸⁶D mutation co-IP'd both γ -tubulin and Grip75-GFP better than Cnn-T-N or Cnn-C Δ^{1-77} -N fragments, and fragments containing both the T²⁷E and S¹⁸⁶D mutations were extremely efficient at co-IPing γ -TuRCs (the mutations appear to be synergistic). Thus, while there are differences between the assays, and perhaps other phosphorylation sites are involved, collectively the data supports a role for T²⁷ and S¹⁸⁶ phosphorylation in relieving auto-inhibition and promoting binding to γ -TuRCs.

After agreement with Reviewer 1 and the Editors (via email discussion), we did not include Cnn-C Δ^{1-77} or phospho-mimetic mutations in the analysis for Figure 5 (*in vivo* analysis). The purpose of this Figure was not to test binding to γ -TuRCs but rather to show that ectopic binding can lead to ectopic cytosolic microtubules and defects during cell division.

The table below summarises the various N-terminal modifications analysed in the different assays and indicates where the data is included in the new manuscript:

Assay	N-terminal modification	Figures
Co-IP assay	Cnn-C, Cnn-T, Cnn-C Δ^{1-77} Cnn-CP ¹ , Cnn-CP ² , Cnn-CP ³ , Cnn-CP ¹⁻³ , Cnn-CP ^{1a} , Cnn-CP ^{1b} , Cnn-CT ^{27E} , Cnn-CS ¹⁸⁶ , Cnn-CT ^{27E,S186D} .	Figure 1 Figure 4
γ -tubulin recruitment to scaffolds	Cnn-C, Cnn-T, Cnn-C Δ^{1-77} , Cnn-C Δ^{1-97} Cnn-CT ^{27E} , Cnn-CS ^{173D} , Cnn-CS ¹⁸⁶ , Cnn-CT ^{27E,S186D} .	Figure 1 Figure 4
Grip75 recruitment to scaffolds	Cnn-C, Cnn-T, Cnn-C Δ^{1-77} Cnn-CT ^{27E} , Cnn-CS ¹⁸⁶ , Cnn-CT ^{27E,S186D} .	Figure 2 Figure 4
Microtubule organisation at scaffolds	Cnn-C, Cnn-T, Cnn-C Δ^{1-77} Cnn-CT ^{27E,S186D} .	Figure 3 Figure 4

2. Figure 3: The amounts of Cnn-C at the spindle poles look smaller than those of Cnn-T. It may be worth to compare the aster intensities of Cnn-C and Cnn-T with their spindle pole intensities.

The Reviewer is right to question the relationship between scaffold size and microtubule organisation ability. The size of the scaffolds formed after mRNA injection is variable. We have re-acquired and re-analysed the data and in order to account for an effect of scaffold size we analysed only those eggs that contained scaffolds of a similar size across the different conditions. While measuring aster intensities is in theory a good idea, is it unfortunately not straightforward. For example, an aster with long but sparse microtubules cannot easily be compared quantitatively with an aster with short but dense microtubules, especially because it is not possible to resolve individual microtubules in this assay. We therefore analysed the data by blind categorisation of the eggs, as we had initially done. The results still clearly show a difference between scaffold types (Figure 3D and Figure 4I).

3. Figure 5E: The authors may explain why a half of the wild type embryos had defects. This point was also raised by Reviewer 2, point 3. We apologise for the confusion caused, which was primarily a consequence of the way the embryos had been originally characterised. We had originally categorised embryos into four groups: mild, moderate, severe, or no defects, Within the “mild defect” category, we had included embryos where only one or two mitotic figures (out of the many present in the embryo) had even a mild defect, which is actually perfectly normal even for wild-type embryos. These very mild defects do not affect embryonic development, as small numbers of defective nuclei can be accounted for (they fall into the centre of the embryo prior to cellularisation). Given the confusion that the original categorisation has caused, we have now repeated the analysis and included any such embryos in the “normal” category. The new analysis was again carried out blind and by a different lab member. The new data, presented in the new Figure 6G, shows that almost 80% of wild-type embryos are normal. It also reveals a clearer difference between pUbc-Cnn-C and pUbc-Cnn-C^T embryos.

4. Figure S3: The putative CAI sequence may be compared with the *C. elegans* and human homologues. Furthermore, experiments in Figure 4A-D may be performed with human CDK5RAP2.

We have now included a new alignment comparing the N-terminal regions of *D. melanogaster* Cnn, human CDK5RAP2, *C. elegans* SPD-5, *A. aegypti* Cnn, *S. pombe* Mto1 and Pcp1, and *S. cerevisiae* SPC110 and SPC72 (Figure S3B). The different homologues all have sequence upstream of the CM1 domain but only the CM1 domain is sequence-conserved. Nevertheless, the alignment is useful as it highlights the relative positions of phosphorylation sites that promote γ -TuRC binding in humans, *C. elegans*, and *S. cerevisiae* which have similar positions to T²⁷ and the new S¹⁸⁶ site in *Drosophila* that we have identified since the original submission.

As requested, we performed co-IP experiments with N-terminal fragments of human CDK5RAP2 and the results are presented in Figure 8. These experiments show that CDK5RAP2 is also auto-inhibited from binding γ -TuRCs, but that it is the region downstream, not upstream, of the CM1 domain that inhibits γ -TuRC binding. Nevertheless, while the details may vary, auto-inhibition of binding to γ -TuRCs appears to be a conserved feature of CM1 domain proteins and we feel this result strengthens the main message of the paper.

5. Since CAI inhibits γ -TuRC binding to CM1, it may physically interact with CM1. The physical interaction could be regulated by phosphorylations at specific residues. This prediction may be examined with the GST pulldown and/or coimmunoprecipitation assays with the CAI and CM1 fragments.

We agree with the Reviewer that an explanation for how inhibition occurs is direct binding between the CAI and CM1 domains that is then released by phosphorylation. Based on the Reviewer's suggestion, we purified CAI domain fragments (including the whole CAI domain or just the α -helix within the CAI domain) and fragments containing the CM1 domain, but initial experiments failed to show an interaction. It is possible that this was due to the relatively low protein concentration that we obtained for the CAI domain fragments, or that the individual domains do not fold properly, or that we have not yet found the appropriate conditions required for an interaction *in vitro*. An interaction would presumably be more likely when the two domains are physically linked to each other, as they normally are *in vivo*. This is certainly something we want to pursue further, but this will take more time than is reasonable for this revision. We have therefore toned down our language regarding our idea of a fold-back mechanism throughout the manuscript (see examples listed at the bottom of this letter). We decided not to discuss our preliminary interaction tests within the manuscript as it is still possible that an interaction could occur under appropriate conditions.

Reviewer #2 (Comments to the Authors (Required)):
Major Comments:

- 1) Central to the authors' model, the CAI inhibits the ability of Cnn to interact with g-tubulin complexes. This conclusion is clear from both the *in vivo* and *in vitro* data. Similarly, it is clear from the data that the "uninhibited" form of Cnn can drive assembly of microtubules *in vivo*. Given that condensates of the Cnn ortholog SPD-5 have been shown to be able to induce microtubule growth independent of g-TuRC, I think it is crucial to show that the ability of Cnn condensates to build microtubules in this *in vivo* assay is g-tubulin dependent. This could be done through g-tubulin depletion experiments combined with the expression of CnnC or CnnT in eggs. The Editors felt that this experiment was not essential: *"the comments about condensates (reviewer 2 point 1) as well as the concerns regarding results in embryos all seem less essential to address experimentally"*. We assume that this is because the result is already suggested by our scaffold analysis. Cnn-C and Cnn-T scaffolds would presumably have organised microtubules equally well if the microtubules were generated solely by concentrating α/β -tubulin, but we find that Cnn-T scaffolds organise microtubules much better than Cnn-C scaffolds. We conclude that this is most likely due to the difference in γ -TuRC recruitment. Of course, it is possible that the short testes-specific N-terminal region (found only in Cnn-T scaffolds) can somehow lead to an increased ability of the scaffolds to concentrate tubulin, but our new finding that Cnn-C ^{Δ 1-77} scaffolds organise microtubules nearly as well as Cnn-T scaffold rules out this possibility. Thus, rather than addressing this experimentally, we have added the following statement to the main text: lines 212-218: *"While it is possible that some microtubules could have been generated independently of γ -TuRCs, a process that occurs by tubulin concentration at C. elegans SPD-5 condensates formed in vitro (Woodruff et al., 2017), the increased microtubule organising capacity at Cnn-T and Cnn-C ^{Δ 1-77} scaffolds (high γ -TuRC recruitment) compared to at Cnn-C scaffolds (low γ -TuRC recruitment) suggests that γ -TuRC-mediated microtubule nucleation/organisation is the predominant factor at these Cnn scaffolds."*

Another key point in the authors' model is that phosphorylation of the CAI by Polo relieves the inhibition of this region and allows for g-tubulin complex binding. This connection is weak at best. The authors demonstrate that phosphomimetics have increased g-tubulin binding *in vitro*, however, they show neither that these phosphorylation sites matter *in vivo* nor that they are regulated by Polo. Potential experiments to test either of these claims are referenced in the Discussion, but the data is not shown and despite having negative data the authors do not change the claims of their model. The language needs to be significantly softened throughout and experiments should be presented as Supplemental data to make it clear that the current data do

not support this part of the model.

We agree with the Reviewer that the connection between phosphorylation of the CAI domain by Polo and the release of inhibition was weak, and we have now removed reference to a potential role for Polo in the revised manuscript. This comment also pushed us to establish better whether phosphorylation really does help relieve auto-inhibition. As a means to assess a role for phosphorylation *in vivo* relatively quickly, we used the scaffold assay to test the effect of phospho-mimicking T²⁷ as well as other putative sites within cells. Encouragingly, we found that phospho-mimicking T²⁷ and another site that we identified during the revision, S¹⁸⁶, increased the recruitment of γ -tubulin to scaffolds (Figure 4G). Recruitment of γ -tubulin was not as robust as the recruitment seen at Cnn-T and Cnn-C ^{Δ 1-77} scaffolds, and this appeared to be because the scaffolds with N-terminal phospho-mimetic mutations seem to recruit only γ -TuSCs (they did not recruit Grip75^{GCP4}-GFP – Figure 4H). However, as mentioned above, phospho-mimicking S¹⁸⁶ had a very strong effect in the *in vitro* co-IP assay and led to binding of γ -TuRCs, not just γ -TuSCs (as Grip75^{GCP4}-GFP, as well as γ -tubulin, could be co-IP'd – Figure 4E). Phospho-mimicking both T²⁷ and S¹⁸⁶ resulted in extremely efficient binding to γ -TuRCs (much higher than when using the “high binding” Cnn-T-N fragments - Figure 4E).

We cannot yet explain the difference between the assays – perhaps the mimetic mutations function better *in vitro* – or perhaps other phosphorylation sites are required in full-length Cnn (used in scaffold assay) as opposed to N-terminal fragments (used in the co-IP assay). Nevertheless, the data collectively suggest that these sites are important and appear to function *in vivo*. While we feel we have made progress, we would agree that we still do not have final proof that these sites are phosphorylated and that they definitely have a role *in vivo* (we would need to generate fly lines with point mutations, which we feel is beyond the scope of this revision). We have therefore ensured that our language regarding phosphorylation has been toned down throughout the manuscript (see examples listed at the bottom of this letter) and certainly plan to investigate this further in the near future.

- 2) The authors claim that the ability to regulate microtubule growth in time and space matters to the cell and organism. While the sperm defects from CnnT overexpression are clear, the defects presented in the embryo are not. In particular, in Figure 5E I find it alarming that the wild-type control in this experiment has such a significant proportion of mitotic defects.

Please see our response to the same point made by Reviewer 1, point 3. Briefly, we have re-analysed and re-categorised the embryos to take account of the fact that it is normal for wild-type syncytial embryos to have defects in a few mitotic figures. This does not prevent normal development. The new categorisation shows that around 80% of wild-type embryos are normal (Figure 6G).

The fact that the control is so perturb makes me question whether the phenotypes seen in CnnC and CnnT eggs are due to the specific effect of these proteins on the process in question vs. a synthetic interaction with background markers.

Please note that these embryos were fixed and immunostained and so they did not express any “background markers”. We have tried to make this clearer in the main text: “*We did, however, frequently observe defects **in fixed and stained** syncytial embryos from pUbq-Cnn-C^T females (Figure 6D-F) as compared to embryos from pUbq-Cnn-C females (Figure 6A-C).*”

In the discussion, the authors note that spurious cytoplasmic microtubule nucleation “major defects during cell division” (line 503), which is an overstatement based on the control in this case. Is this indeed a background effect?

We hope that the statement now makes more sense given that the majority of wild-type embryos were actually normal and that no background markers were expressed. Indeed, the strongest defects, where the entire embryo is massively perturbed, were only observed in pUbq-Cnn-T embryos, and these defects occur even though pUbq-Cnn-T expression is very low in embryos

(see Figure 5C). Given this, and the clear defects observed within testes, we hope the Reviewer agrees that it is valid to say that ectopic cytosolic microtubules appear to lead to major cell division defects.

Are the same effects seen in CnnC or CnnT eggs that have been fixed and stained instead of expressing other endogenously tagged proteins?

We apologise for not making this clearer. As noted above, these embryos (and the testes) were fixed and stained and did not express any tagged proteins.

Minor comments:

-Line 126: missing character

Thanks for pointing this out – it seems the conversion to PDF introduced this error – this section has now been rephrased and so the error is hopefully now absent.

-Figure 5F is not referenced in the main Results section

The embryo pictures are now included in new Figure 6 and are referenced in the main results section (lines 340-353).

-Define Nebenkern

We now say: *“When meiosis progresses normally, the 64 round spermatids cells within the resulting cyst all contain a similarly sized phase-light nucleus and phase-dark nebenkern (**which is an accumulation of mitochondria that were segregated during meiosis**)”.*

-Line 489: nucleat[ed]

The text in this section has now been modified

Reviewer #3 (Comments to the Authors (Required)):

- 1) The main message of the manuscript is on the autoinhibitory function of the Cnn amino-terminus. The data on hatching and embryonic defects in *Drosophila* are of lower priority, as long as the cellular mechanisms aren't documented more clearly. I think the authors are skipping steps here. Whereas it is well documented that artificial Cnn-T scaffolds attract gamma-TuRCs and induce microtubule asters (Figures 2 and 3), it would be helpful to have clearer pictures of mitotic spindles and Cnn-CT-induced spindle defects in embryos, and well-resolved images on the formation of supernumerary microtubule-organizing centres (Figure 5). The immunofluorescence images in Figures 5F and 5I are too small and too fuzzy to make a strong point; and Figure S5 should show control cells (Cnn-C or wild type) for comparison.

We agree that the images of embryo defects were too small in the original manuscript. We have increased their size and presented them in new Figure 6. We have also removed the overlaid text to make the defects easier to observe, and added another example of defects at telophase (Figure 6F). As discussed in the responses to Reviewers 1 and 2 we have also re-characterised the embryos, which has made a clearer quantitative distinction between pUbq-Cnn-C^T and wild-type/pUbq-Cnn-C embryos (Figure 6G). As for the spermatocyte images in original Figure S5, we have now included a complementary pUbq-Cnn-C image for each pUbq-Cnn-C^T image (i.e. a cell at the same developmental stage), as requested (Figure 7D; Figure S5). Note that these are all newly acquired images from new samples - we present these chronologically at matched developmental stages.

- 2) The autoinhibitory action of the Cnn-C amino-terminal region should be tested more rigorously, with additional experiments: can the authors produce protein fragments in bacteria containing P1, P2, P1+2, CM1+P3, and test directly whether the P1-containing

fragments bind to the CM1 region? This would be important as direct proof of their model in Figure 4A, suggesting the folding-back of P1+P2 onto Cnn.

This is essentially the same important point as made by Reviewer 1, point 5. As discussed in the response to Reviewer 1, we have purified separate CAI domain and CM1-domain containing fragments but initial experiments failed to demonstrate an interaction in this context. While it would have been nice to prove such an interaction exists, we expect that optimising the conditions will take longer than is reasonable for this current revision. We have therefore toned down our language throughout the text to make it clear that the fold-back model has not yet been proved (see examples listed at the bottom of this letter).

Other points:

- 3) Lines 199-216 and Figure 2: The S values for Grip75, Grip 128, and Grip163 vary from gamma-Tubulin37C, and the authors speculate that this might be due to the lower abundance of these three GCPs within the gamma-TuRC. But why are the values for Grip128 and Grip163 so different from each other, although both proteins are present in the gamma-TuRC at the same stoichiometry?

We suspect that the difference occurred due to noise in the data, given that Grip128 and Grip163 are low stoichiometry proteins, although one cannot rule out that not all γ -TuRCs contain Grip163 or Grip128. In any case, this data is no longer present in the revised manuscript because we have focussed on analysing Grip75-GFP recruitment (as this is the better marker of γ -TuRCs and analysing all 3 proteins with all of the new scaffold types would have required an unrealistic amount of work).

- 4) I think that Figure 3D and the corresponding comments in the 'results' section are unnecessary and are distracting from the main message of the manuscript (as long as the spindle defects in cells/embryos aren't documented more clearly).

We appreciate that our comments on spindle structures forming between scaffolds is somewhat distracting from the main message, but the point of this was to emphasise how the microtubules organised by Cnn-T scaffolds are very likely to be dynamic. In our view, this makes it more likely that the microtubules were nucleated from the scaffolds, rather than having been attached post-nucleation. We therefore feel showing these spindle structures is relevant, but we have now cut down the corresponding comments in the results section (see lines 222-227). We have also more clearly documented the spindle defects in embryos and spermatocytes, as requested.

Below is a list of the other changes we have made to the Revised version

- a. We increased the numbers of mCherry-Jupiter eggs injected with GFP-Cnn-T-N (or water as a control) and quantified the percentage of eggs that contained cytosolic microtubules ~2h after injection. This data is reported on lines 327-331 and a new Video S5 has been included.
- b. In order to tone down our conclusions and soften the language in relation to phospho-regulation and the potential folding-back of the CAI domain, the following changes were made:
 - I. Title has changed from: *“Phospho-regulated auto-inhibition of Cnn controls microtubule nucleation during cell division”* to *“Auto-inhibition of Cnn binding to γ -TuRCs prevents ectopic microtubule nucleation and cell division defects”*
 - II. We have modified the abstract to tone down our findings regarding phosphorylation: *“Robust binding occurs after removal of the CAI domain or with the addition of phospho-mimetic mutations, suggesting that phosphorylation helps relieve inhibition.”* and to remove the old summarising sentence: *“We propose that*

- the CAI domain folds back to sterically inhibit the CM1 domain, and that this auto-inhibition is relieved by phosphorylation that occurs specifically at centrosomes.*"
- III. At the start of the results section, when citing our previous paper, we have removed reference to our proposal that folding back of the N-terminal region might mediate auto-inhibition. Previously: *"we had hypothesised that the larger extreme N-terminal region of Cnn-C may fold back and auto-inhibit the CM1 domain, restricting its ability to bind γ -TuRCs"*; we now say *"We had hypothesised that the larger extreme N-terminal region of Cnn-C may auto-inhibit the CM1 domain, restricting its ability to bind γ -TuRCs"*.
 - IV. Figure 4 title has changed from *"Phosphorylation of the CAI domain relieves auto-inhibition to allow binding to γ -tubulin complexes"* to *"Phospho-mimetic mutations within the CAI domain and downstream of the CM1 domain promote binding to γ -tubulin complexes."*
 - V. In Figure 4a, we have changed the diagram so that the N-terminal region is not shown in a folded state – we only indicate with an arrow and associated question mark that the CAI may fold back to block the CM1 domain.
 - VI. We have changed a results section title from *"Phosphorylation of a conserved helix within the CAI domain, including a conserved Polo kinase site, helps relieve auto-inhibition"* to *"Phospho-mimetic mutations help relieve CAI domain mediated auto-inhibition"*
 - VII. At the start of the phospho-regulation results section, we removed the sentence: *"Given the similar length of the CAI and CM1 domains, we speculated that the CAI domain may fold back over and inhibit the CM1 domain (Figure 4A)".* We now only talk about how phosphorylation is a good candidate for how inhibition could be relieved.
 - VIII. At the end of the phospho-regulation section we make it clear that the mechanism is not yet fully understood: *"Thus, while there are some differences between the scaffold assay and the co-IP assay, the data collectively **suggest** that phosphorylation at T²⁷ and in particular at S^{186D} help to relieve CAI domain auto-inhibition and promote the binding of Cnn-C to γ -TuRCs."*
 - IX. At the start of the Discussion section, we have removed: *"We propose that the CAI domain folds back to sterically inhibit the CM1 domain, and that this inhibition is relieved only once Cnn-C is recruited to centrosomes and phosphorylated at sites within the CAI domain, including the predicted Polo site T²⁷".*
 - X. We now more tentatively discuss how the CAI domain may fold back to inhibit the CM1 domain in the discussion, highlighting that this idea has not yet been proved: *"In future, it will be important to understand how the CAI domain inhibits the CM1 domain. We previously postulated that the extreme N-terminal region of Cnn-C (i.e. the CAI domain) might fold back and sterically inhibit the CM1 domain (Tovey et al., 2018). Our data is consistent with this possibility and, in our view, this is the most likely explanation. A similar mechanism has also been proposed in C. elegans (Ohta et al., 2021). Nevertheless, there are alternative possibilities, including that the CAI domain could recruit another protein that interferes with CM1 domain binding. In any case, it will be interesting to compare how auto-inhibition is achieved in different homologues, especially given that the region downstream, not upstream, of the CM1 domain appears to mediate inhibition in human CDK5RAP2."*

Once again, we thank the Reviewers for their carefully considered comments. We hope they agree that the comments have improved the paper and that it is now ready for publication.

With kind regards,
Paul Conduit

April 21, 2021

RE: JCB Manuscript #202010020R

Dr. Paul T Conduit
Institut Jacques Monod
15 rue Hélène Brion
Paris 75205
France

Dear Dr. Conduit:

Thank you for submitting your revised manuscript entitled "Auto-inhibition of Cnn binding to γ -TuRCs prevents ectopic MT nucleation and cell division defects". We would be happy to publish your paper in JCB pending final revisions necessary to meet our formatting guidelines (see details below). In your final revision, please be sure to address reviewer #2's important final concerns.

A. MANUSCRIPT ORGANIZATION AND FORMATTING:

Full guidelines are available on our Instructions for Authors page, <https://jcb.rupress.org/submission-guidelines#revised>. **Submission of a paper that does not conform to JCB guidelines will delay the acceptance of your manuscript.**

- 1) Text limits: Character count for Articles is < 40,000, not including spaces. Count includes title page, abstract, introduction, results, discussion, acknowledgments, and figure legends. Count does not include materials and methods, references, tables, or supplemental legends.
- 2) Figures limits: Articles may have up to 10 main text figures.
- 3) Figure formatting: Scale bars must be present on all microscopy images, including inset magnifications. * Molecular weight or nucleic acid size markers must be included on all gel electrophoresis. *
- 4) Statistical analysis: Error bars on graphic representations of numerical data must be clearly described in the figure legend. The number of independent data points (n) represented in a graph must be indicated in the legend. Statistical methods should be explained in full in the materials and methods. For figures presenting pooled data the statistical measure should be defined in the figure legends. Please also be sure to indicate the statistical tests used in each of your experiments (either in the figure legend itself or in a separate methods section) as well as the parameters of the test (for example, if you ran a t-test, please indicate if it was one- or two-sided, etc.). Also, if you used parametric tests, please indicate if the data distribution was tested for normality (and if so, how). If not, you must state something to the effect that "Data distribution was assumed to be normal but this was not formally tested."

- 5) Abstract and title: The abstract should be no longer than 160 words and should communicate the significance of the paper for a general audience. The title should be less than 100 characters including spaces. Make the title concise but accessible to a general readership.
- 6) Materials and methods: Should be comprehensive and not simply reference a previous publication for details on how an experiment was performed. Please provide full descriptions in the text for readers who may not have access to referenced manuscripts.
- 7) * Please be sure to provide the sequences for all of your primers/oligos and RNAi constructs in the materials and methods. You must also indicate in the methods the source, species, and catalog numbers (where appropriate) for all of your antibodies. Please also indicate the acquisition and quantification methods for immunoblotting/western blots. *
- 8) Microscope image acquisition: The following information must be provided about the acquisition and processing of images:
- Make and model of microscope
 - Type, magnification, and numerical aperture of the objective lenses
 - Temperature
 - Imaging medium
 - Fluorochromes
 - Camera make and model
 - Acquisition software
 - Any software used for image processing subsequent to data acquisition. Please include details and types of operations involved (e.g., type of deconvolution, 3D reconstitutions, surface or volume rendering, gamma adjustments, etc.).
- 9) References: There is no limit to the number of references cited in a manuscript. References should be cited parenthetically in the text by author and year of publication. Abbreviate the names of journals according to PubMed.
- 10) Supplemental materials: There are strict limits on the allowable amount of supplemental data. Articles/Tools may have up to 5 supplemental display items (figures and tables). Please also note that tables, like figures, should be provided as individual, editable files. A summary of all supplemental material should appear at the end of the Materials and methods section.
- 11) eTOC summary: A ~40-50-word summary that describes the context and significance of the findings for a general readership should be included on the title page. The statement should be written in the present tense and refer to the work in the third person.
- 12) Conflict of interest statement: JCB requires inclusion of a statement in the acknowledgements regarding competing financial interests. If no competing financial interests exist, please include the following statement: "The authors declare no competing financial interests." If competing interests are declared, please follow your statement of these competing interests with the following statement: "The authors declare no further competing financial interests."
- 13) ORCID IDs: ORCID IDs are unique identifiers allowing researchers to create a record of their various scholarly contributions in a single place. At resubmission of your final files, please consider providing an ORCID ID for as many contributing authors as possible.
- 14) A separate author contribution section following the Acknowledgments. All authors should be

mentioned and designated by their full names. We encourage use of the CRediT nomenclature.

B. FINAL FILES:

-- High-resolution figure and video files: See our detailed guidelines for preparing your production-ready images, <https://jcb.rupress.org/fig-vid-guidelines>.

Thank you for this interesting contribution, we look forward to publishing your paper in Journal of Cell Biology.

Sincerely,

Tarun Kapoor, PhD
Monitoring Editor

Andrea L. Marat, PhD
Senior Scientific Editor

Journal of Cell Biology

Reviewer #1 (Comments to the Authors (Required)):

The authors revised the manuscript, addressing most of the points that had been raised. I believe that the manuscript is ready to be published in JCB.

Reviewer #2 (Comments to the Authors (Required)):

Tovey et al. resubmit their manuscript showing a novel regulatory region in Cnn that modulates binding to g-TuRC. The new version of the manuscript is much improved, with several added experiments, including identification of a new phosphor-site that appears to also function in regulating Cnn autoinhibition, testing Cnn phosphomutants in the in vivo scaffold assay, better clarification of the effects on introducing uninhibited forms of Cnn in embryos, and a demonstration of conservation of an inhibitory region of the human homolog of Cnn, CDK5RAP2. The authors also removed experiments referencing a role for Polo and instead focused the manuscript only on the Cnn side of things and the consequences of its uninhibition in vivo, which I think makes for a stronger overall message. I am generally satisfied with the resubmission. I have a few minor suggestions for the manuscript and figures that I think will further strengthen the manuscript below:

-Figure 4: The authors see a difference in the behavior of the T27,S186 in in vitro vs. in vivo assays, but in the in vitro one case they are using just the N-terminal fragment and in the in vivo case they are using the full length. Do the authors have any in vitro data with the full length T27, S186 or in vivo data with the N terminal fragment of T27, S186 for a more direct comparison?

-If desired for space, figures could be condensed. For example: Figure 2 could go in Supplement either alone or combined with Figure S2; Figure 6 and 7 could be combined.

-Figure 3 and Figure 6: consider adding an inset with a magnified view of one scaffold (Fig. 3) or a spindle (Fig. 6) in each condition. The images in figure 6 are less clear than those included in the previous version for this point.

-I personally do not think that possessives (e.g. Cnn's, line 270) have a place in scientific writing. Particular egregious: C. elegans' SPD-5 (line 272)

-Line 329: can use consistent nomenclature here, i.e. Cnn-T-N

-Video 5 seems like an afterthought and not incorporated with the results in its current position in the text. It would also be nice to have stills of this in the main or Supplemental figure

-what is the point of showing both Figure 8B and C? Figure 8C is not referenced in the text.

-Line 453-454: Unclear what is meant here by "to rely on phosphorylation for binding". To what? g-TuRC? The nebenkern? Both?

-In the discussion of the regulation of Cnn-T and its regulation of g-TuRC in the sperm, can you rule out that the Cnn-T isoform doesn't have autoinhibitory domains in its C-terminus?

-I am confused by the sentence on line 454: Binding and potential activation of g-TuRCs...is not detrimental to sperm development" when in fact expression of Cnn-CT in the testes was incredibly detrimental and significantly reduced fertility. Later defects could be masked by meiotic defects

already shown.

Dear Reviewer 2,

Many thanks for taking the time to thoroughly Review and check our manuscript. Your comments have really helped us improve the paper. Below we address your final concerns.

Reviewer 2 final comments are in grey, our responses in black, and quoted text in orange.

-Figure 4: The authors see a difference in the behavior of the T27,S186 in in vitro vs. in vivo assays, but in the in vitro one case they are using just the N-terminal fragment and in the in vivo case they are using the full length. Do the authors have any in vitro data with the full length T27, S186 or in vivo data with the N terminal fragment of T27, S186 for a more direct comparison?

Unfortunately, we do not have any such data. In future, we can consider trying to purify full-length Cnn, possibly using insect cells. We can also try linking N-terminal regions of Cnn to GBP and forcing their recruitment to GFP-Cnn scaffolds that are otherwise incapable of recruiting γ -TuRCs.

-If desired for space, figures could be condensed. For example: Figure 2 could go in Supplement either alone or combined with Figure S2; Figure 6 and 7 could be combined. Should the Journal want us to reduce the Figure count, combining Figure 2 into Figure S2 is a reasonable suggestion. However, showing Grip75-GFP recruitment to various scaffold types was one of the major comments from Reviewer 1 and so we prefer to leave it as a main Figure if possible. As for Figure 6 and 7, in the original submission data from new Figures 6 and 7 were presented in only one Figure; however, Reviewer 3 felt that the images of embryos were too small. We have therefore left Figures 6 and 7 separate to allow for large images of embryos and spermatocytes.

-Figure 3 and Figure 6: consider adding an inset with a magnified view of one scaffold (Fig. 3) or a spindle (Fig. 6) in each condition. The images in figure 6 are less clear than those included in the previous version for this point.

We have added insets for the scaffolds in Figure 3, as suggested. For Figure 6, however, we feel this is less appropriate because it is difficult to truly reflect the overall disruption to the syncytium using an image of a single mitotic figure. We are surprised that the Reviewer found these images less clear than those in the original submission, because these are the same images (along with two additional images), only larger.

-I personally do not think that possessives (e.g. Cnn's, line 270) have a place in scientific writing. Particular egregious: *C. elegans*' SPD-5 (line 272)

Agreed. Both now corrected, including another one at the end of the results section.

-Line 329: can use consistent nomenclature here, i.e. Cnn-T-N
True, now corrected.

-Video 5 seems like an afterthought and not incorporated with the results in its current position in the text. It would also be nice to have stills of this in the main or Supplemental figure

We have now moved the description of the data in Video 5 back to its previous position in the original submission, just after the description of embryo defects. We have also added still images in Figure 6 (Figure 6H).

-what is the point of showing both Figure 8B and C? Figure 8C is not referenced in the text. We apologise for this confusion, we had made a typo when citing this figure, saying Figure 8A,B instead of Figure 8B,C. Fragments essential for our conclusions were tested in Figure

8C (aa1-100 and aa51-210) that were not tested in Figure 8B. We have therefore kept Figure 8C and corrected the typo.

-Line 453-454: Unclear what is meant here by "to rely on phosphorylation for binding". To what? γ -TuRC? The nebenkern? Both?

This should be "to γ -TuRCs"; now added.

-In the discussion of the regulation of Cnn-T and its regulation of γ -TuRC in the sperm, can you rule out that the Cnn-T isoform doesn't have autoinhibitory domains in its C-terminus? It is true that we cannot completely rule out that the C-terminal region of Cnn-T can auto-inhibit γ -TuRC binding and that this putative auto-inhibition requires phosphorylation to be relieved. We have now added this possibility into the discussion: *"While we cannot rule out that Cnn-T contains autoinhibitory domains in its C-terminal region, we have shown that the N-terminal region of Cnn-T can bind efficiently to γ -TuRCs in the apparent absence of any upstream regulatory events."*

-I am confused by the sentence on line 454: Binding and potential activation of γ -TuRCs...is not detrimental to sperm development" when in fact expression of Cnn-CT in the testes was incredibly detrimental and significantly reduced fertility. Later defects could be masked by meiotic defects already shown.

We apologise for this confusion, which was due to a lack of clear writing on our part. We have now re-worded this paragraph to better explain our thoughts (see new paragraph pasted in orange below). The key to our thinking is that many testes-specific genes are predominantly expressed post-meiotically. We presume this is true for Cnn-T, particularly because microtubule nucleation from mitochondria has not been reported in mitotic or meiotic cells within the testes, or in any dividing cells to our knowledge. Should Cnn-T expression be restricted until after meiosis, this would mean that binding and activation of cytosolic γ -TuRCs by Cnn-T would occur only in developing sperm cells, where cytosolic microtubule nucleation is presumably not a problem for sperm development (as they have a shrinking cytosol and do not need to form a spindle).

"Importantly, our data also highlights differences in how binding between CM1 domain proteins and γ -TuRCs is regulated within different cell types and at different MTOCs. Testes-specific Cnn-T isoforms lack the CAI domain and recruit γ -TuRCs to mitochondria in developing sperm cells (Chen et al., 2017). While we cannot rule out that Cnn-T contains autoinhibitory domains in its C-terminal region, we have shown that the N-terminal region of Cnn-T can bind efficiently to γ -TuRCs in the apparent absence of any upstream regulatory events. The surface of mitochondria presumably lacks the kinases that regulate Cnn-C at centrosomes and so it would seem appropriate that Cnn-T can bind γ -TuRCs in the absence of phosphorylation. This would make most sense should Cnn-T isoforms be predominantly expressed post-meiotically, which is common for testes-specific genes (White-Cooper, 2012), as their binding and activation of cytosolic γ -TuRCs may perturb dividing cells but presumably would not perturb developing sperm cells, which have a shrinking cytosol and no need to form a spindle."

We hope that Reviewer 2 is now satisfied and agrees that the paper is ready for publication

With kind regards,

Paul